# Autophagy-related protein Atg11 is essential for microtubule-mediated chromosome segregation

**Md. Hashim Reza[1], Rashi Aggarwal[1◉], Jigyasa Verma[1¤◉], Nitesh Kumar Podh[2], Ratul Chowdhury[3], Gunjan Mehta[2], Ravi Manjithaya[4], Kaustuv Sanyal** [1,5]*

**1** Molecular Mycology Laboratory, Molecular Biology and Genetics Unit, Jawaharlal Nehru Centre for Advanced Scientific Research, Bengaluru, Karnataka, India, **2** Laboratory of Chromosome Dynamics and Gene Regulation, Department of Biotechnology, Indian Institute of Technology Hyderabad, Kandi, Sangareddy, Telangana, India, **3** Department of Chemical and Biological Engineering, Iowa State University, Ames, Iowa, United States of America, **4** Autophagy Laboratory, Molecular Biology and Genetics Unit, Jawaharlal Nehru Centre for Advanced Scientific Research, Jakkur, Bengaluru, Karnataka, India, **5** Department of Biological Sciences, Bose Institute, Unified Academic Campus, Kolkata, West Bengal, India

◉ These authors contributed equally to this work.
¤Current address: Department of Cellular and Physiological Sciences, University of British Columbia, Vancouver, Canada
* sanyal@jncasr.ac.in

## Abstract

Emerging studies hint at the roles of autophagy-related proteins in various cellular processes. To understand if autophagy-related proteins influence genome stability, we sought to examine a cohort of 35 autophagy mutants in *Saccharomyces cerevisiae.* We observe cells lacking Atg11 show poor mitotic stability of minichromosomes. Single-molecule tracking assays and live cell microscopy reveal that Atg11 molecules dynamically localize to the spindle pole bodies (SPBs) in a microtubule (MT)-dependent manner. Loss of Atg11 leads to a delayed cell cycle progression. Such cells accumulate at metaphase at an elevated temperature that is relieved when the spindle assembly checkpoint (SAC) is inactivated. Indeed, *atg11Δ* cells have stabilized securin levels, that prevent anaphase onset. Ipl1-mediated activation of SAC also confirms that *atg11Δ* mutants are defective in chromosome biorientation. Atg11 functions in the Kar9-dependent spindle positioning pathway. Stabilized Clb4 levels in *atg11Δ* cells suggest that Atg11 maintains Kar9 asymmetry by facilitating proper dynamic instability of astral microtubules (aMTs). Loss of Spc72 asymmetry contributes to non-random SPB inheritance in *atg11Δ* cells. Overall, this study uncovers an essential non-canonical role of Atg11 in the MT-mediated process of chromosome segregation.

## Introduction

Dynamic interactions between spindle microtubules (MTs) and chromosomes facilitate precise segregation of the genetic material during the cell cycle. Comprising of α/β-tubulin dimers, MTs display inherent polarity, with the minus end proximal to the spindle poles, while

**Data availability statement:** All relevant data are within the paper and its Supporting information files.

**Funding:** Financial support from the Department of Biotechnology, Ministry of Science and Technology, India under the DBT-RA (http://ra.dbtindia.gov.in/) Program in Biotechnology and Life Sciences is gratefully acknowledged by MHR (DBT/2020/January/58). The award of the JC Bose Fellowship (https://serbonline.in/SERB/jcbose_fellowship?HomePage=New) (JCB/2020/000021) from the Science and Engineering Research Board (SERB) and JNCASR intramural funding support to KS is acknowledged. This work was also supported by the Science and Engineering Research Board (SERB) grant (https://serbonline.in/SERB/HomePage#funding) (CRG/2019/004892) to RM. JV was supported by intramural financial support from JNCASR. RA is supported by intramural financial support from JNCASR. GM lab acknowledges the National Facility for Single-Molecule and Super-Resolution Imaging, funded by the Department of Biotechnology, Ministry of Science and Technology, India under the DBT-SAHAJ (file no. BT/INF/22/SP53103/2024) (https://www.dbtindia.gov.in/sahaj-0) at IIT Hyderabad. NKP acknowledges the Prime Minister's Research Fellowship (https://www.pmrf.in/) (PMRF ID: 2001700) for the financial support. The funders had no role in the study design, data collection and analysis, decision to publish, or preparation of the manuscript.

**Competing interests:** The authors have declared that no competing interests exist.

**Abbreviations:** aMTs, astral microtubules; APC/C, anaphase-promoting complex/cyclosome; *atg*, autophagy-related genes; BiFC, bimolecular fluorescence complementation; CDF, cumulative distribution function; CDK, cyclin-dependent kinase; dSPB, daughter-bound SPB; FEAR, Cdc Fourteen early anaphase release; FRAP, fluorescence recovery after photobleaching; HTL, HaloTag Ligand; MAPs, MT-associated proteins; MT, microtubule; PAS, pre-autophagosomal structure; SAC, spindle assembly checkpoint; SMT, single-molecule tracking; SPBs, spindle pole bodies; TBZ, thiabendazole; Y2H, yeast two-hybrid.

the more dynamic plus end interacts with the chromosomes at the kinetochore. *Saccharomyces cerevisiae* and related budding yeast species undergo closed mitosis with an intact nuclear envelope throughout the cell cycle [1–5]. Spindle pole bodies (SPBs), the microtubule organizing centers, in these organisms, remain embedded in the nuclear envelope throughout the cell cycle [6] and emanate both nuclear and astral microtubules (aMTs). Some of the nuclear MTs attach to kinetochores and facilitate the poleward movement of sister chromatids required for the precise segregation of chromosomes in progeny daughter cells. The spindle assembly checkpoint (SAC), which detects any defects in kinetochore–MT attachments or lack of tension at kinetochores, halts the cell cycle at metaphase until all errors are corrected and prevents any premature separation of sister chromatids [7, 8]. The SAC inhibits the activity of anaphase-promoting complex/cyclosome (APC/C), a ubiquitin ligase required for the degradation of securin and subsequent release of separase, that is necessary for cohesin cleavage and sister chromatid separation [9]. The Ipl1-Aurora B kinase is an evolutionarily conserved important regulator of both kinetochore–MT interactions and the SAC response to tension defects. Ipl1 is known to strengthen SAC activation by destabilizing improper kinetochore–MT attachments [10–12]. Ipl1 carries out this function by phosphorylating outer kinetochore proteins which in turn prevent interactions with the MT, generating unattached kinetochores and thereby activating SAC [13–18]. The tension generated by amphitelic attachment spatially separates Ipl1 from its substrates which allows their dephosphorylation and silences the error correction machinery [17, 18].

The positioning of the mitotic spindle along the polarity axis is critical for accurate chromosome segregation in *S. cerevisiae*, which undergoes asymmetric cell division. Spindle positioning is regulated by two functionally redundant pathways: the Kar9 pathway and the dynein pathway. The Kar9 pathway operates at the pre-anaphase stage of mitosis and involves Bim1 (ortholog of EB1) and Kar9, the type V myosin Myo2, and the kinesin-8 family member, protein Kip3 [19–23]. The dynein-dependent pathway, on the other hand, functions at anaphase, representing the second pathway and involves CLIP-170 homolog Bik1, the cortical protein Num1, and the dynactin complex [24, 25]. The cytosolic face of SPBs nucleates aMTs which aid in the alignment and positioning of the mitotic spindle close to the bud neck and along the polarity axis in *S. cerevisiae*. Two pathways ensure asymmetric nucleation of aMTs from the old SPB (daughter-bound SPB [dSPB]) in yeast. The cyclin-dependent kinase (CDK) Cdc28/Cdk1, representing the intrinsic factor, facilitates the biased recruitment of SPB components such as Spc72 and the γ-tubulin complex at the dSPB, while asymmetric localization of Kar9 at the dSPB, represents the extrinsic factor [26–30]. The asymmetric localization of Kar9 at dSPB is regulated by at least three independent pathways: (a) Cdk1-dependent phosphorylation of Kar9 reduces its affinity to Bim1 and Stu2 [28,31,32], (b) sumoylation of Kar9 facilitates its association with dSPB [33], and (c) SAC activation leads to symmetric Kar9 distribution at both the SPBs [33]. Such asymmetry leads to the nucleation of more aMTs from the dSPB necessary for the mitotic spindle positioning along the mother–bud axis and ensures non-random SPB inheritance [21,28,34–38]. Misaligned spindles activate the spindle position checkpoint [39–43], a surveillance mechanism that halts the cell cycle and prevents exit from mitosis by inhibiting the mitotic exit network signaling pathway [44].

The stochastic transitions between growth and shrinkage, called dynamic instability, are exhibited by MTs in the cell [45]. MT dynamic instability is defined by four parameters: frequency of catastrophe (the transition from polymerization to depolymerization), frequency of rescue (the transition from depolymerization to polymerization), and the rates of polymerization and depolymerization [46]. The dynamic instability facilitates MTs to search for the binding sites on kinetochores (the search-capture model) and thereby helps in the initial attachment of chromosomes to the mitotic spindle [47]. It also aids in the interaction of aMTs

with the cell cortex, crucial for spindle positioning and alignment [21]. Proteins regulating MT dynamics have been classified as either MT-stabilizing factors or MT-associated proteins (MAPs) which regulate MT polymerization or MT destabilizing factors which induce catastrophe [46]. An optimal balance of these regulatory factors ensures controlled MT dynamics facilitating proper and timely nuclear division in the cell. Therefore, any perturbations in MT dynamics have wider implications ranging from failure in embryonic development, early aging, and aneuploidy, to cancer [48–52]. Therefore, a detailed investigation of components that preserve MT integrity crucial for (a) positioning of the nucleus, and (b) high-fidelity kinetochore–MT attachment during the cell cycle is imperative.

Autophagy, apart from being a conserved cellular degradation process, is also implicated in cell division under stress conditions [53, 54] and in controlling the integrity of nuclear-envelope-embedded nuclear pore complexes during the cell cycle [55, 56]. A series of emerging evidence highlights the role of autophagy in cell metabolism, growth, aging, and genome stability [53,54,57]. Under stress and starvation conditions, autophagy is induced and involves Atg17-meditated non-selective degradation of cargoes, while Atg11 mediates selective autophagy predominantly under vegetative conditions [58]. To this end, in this study, we screened a collection of autophagy mutants and identified Atg11 as a regulator of high-fidelity chromosome segregation in *S. cerevisiae*. Atg11 is a scaffolding protein, composed of four coiled-coil domains, and is known to homo-dimerize [59]. Atg11 contributes to the pre-autophagosomal structure (PAS) organization and facilitates the linkage of cargo to the vesicle-forming machinery at the PAS [59, 60] *via* recognition of selective autophagy receptors [61]. Atg11 is important for selective autophagy, but non-essential for bulk autophagy [62]. Atg11 interacts with Atg1, the kinase essential for both selective and non-selective autophagy [62]. The modular nature of Atg11 and the numerous established interaction partners place Atg11 at the hub of selective autophagy. Our results uncover the previously unknown role of Atg11 in enabling MT integrity critical for proper kinetochore-MT interactions. We provide evidence that a significant cellular pool of Atg11 is SPB-associated, ensuring proper cell cycle stage-specific dynamics of aMTs vital for nuclear positioning.

## Results

### Atg11 plays a role in high-fidelity chromosome transmission

To understand the role of autophagy protein(s) in the cell cycle in nutrient-rich conditions, we carried out a phenotypic screen to determine the growth fitness of null mutants of autophagy-related genes (*atg*) in *S. cerevisiae* to a MT depolymerizing drug, thiabendazole (TBZ). Among all the 35 *atg* mutants (S1A Fig), *atg11Δ* cells displayed maximum sensitivity to TBZ (Fig 1A), supporting an earlier observation [63]. Transgenic wild-type *ATG11* complemented the TBZ-induced growth defects in *atg11Δ* cells (Fig 1A). Furthermore, *atg11Δ* cells grew as good as wild-type at 14 °C but significantly slower at 37 °C (Fig 1B), a condition at which the rate of depolymerization of MTs is more dominant than the rate of polymerization [64]. These results hint towards a role of Atg11 in MT-dependent processes.

Any defects in MT-dependent processes negatively impact the fidelity of chromosome segregation. Indeed, a strain harboring an extra linear chromosome displayed a significantly higher rate of chromosome loss in *atg11Δ* cells at 30 °C that was further exacerbated at 37 °C (Fig 1C). Similarly, the average mitotic stability of a monocentric circular plasmid was found to be significantly lower in *atg11Δ* cells than in the wild-type but comparable to that of a kinetochore mutant, *ctf19Δ* (Fig 1D). No other *atg* mutants tested displayed any decrease in the mitotic plasmid stability. Mutants of key autophagy proteins such as Atg1, essential for both cytoplasm-to-vacuole targeting (Cvt] and autophagy pathways, or Atg17, required

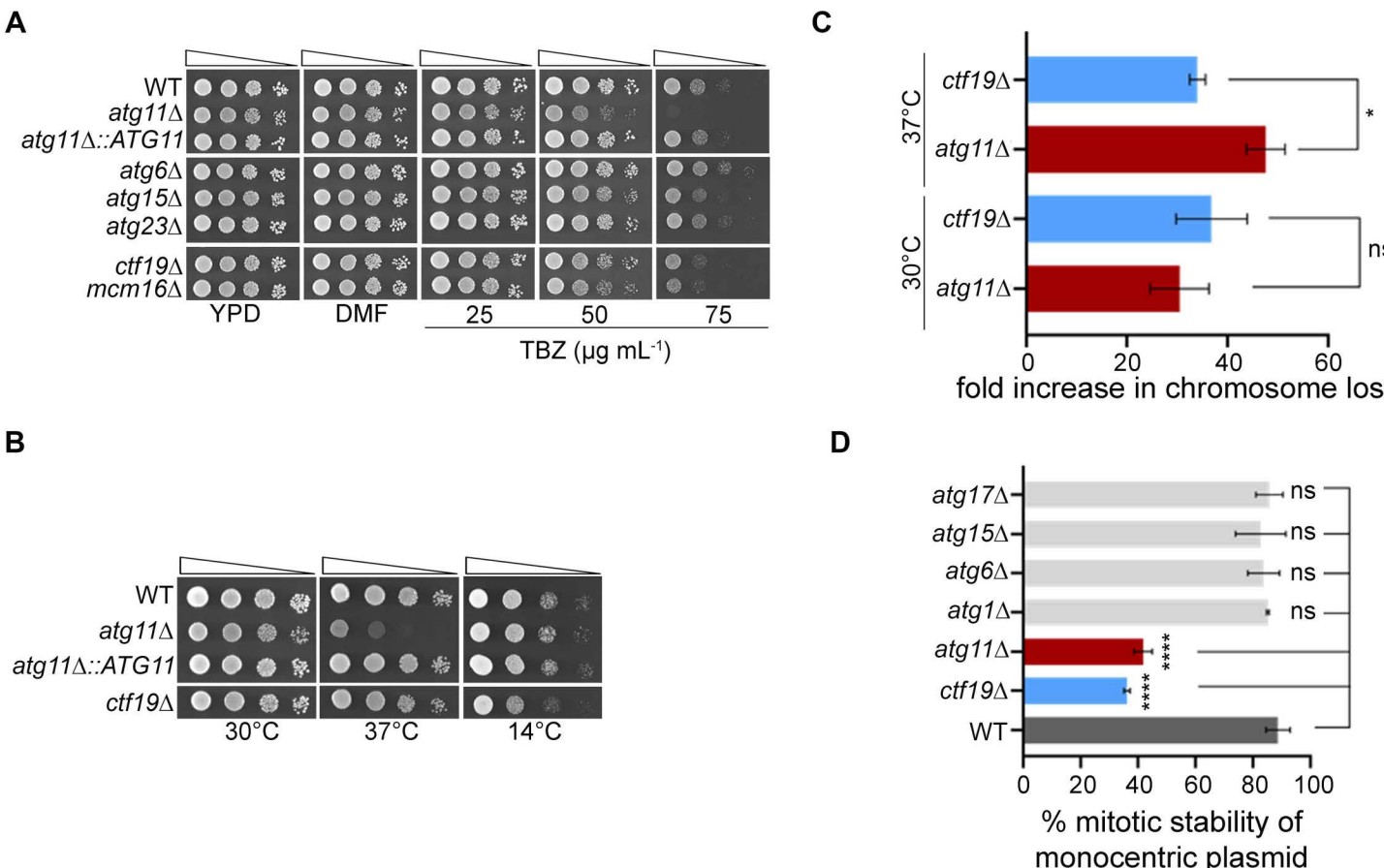

**Fig 1. Cells lacking Atg11 show an increased rate of chromosome loss.** (A) Overnight grown cells of wild-type, null mutants of autophagy-related genes (*atg*), and two kinetochore mutants, *mcm16Δ* and *ctf19Δ*, were 10-fold serially diluted, spotted on YPD and YPD plates containing dimethylformamide (DMF) only or 25, 50, 75 µg mL⁻¹ thiabendazole (TBZ). Plates were photographed after incubation at 30 °C for 48 h. (B) Overnight grown cells of indicated strains were 10-fold serially diluted and spotted on YPD plates. Plates were photographed after incubation at 30 and 37 °C for 48 h, or 14 °C for 7 days. (C) Bar graph showing fold increase in chromosome loss in *atg11Δ* and *ctf19Δ* strains at 30 and 37 °C from three biological replicates. Error bars indicate standard deviation (SD). Statistical analysis was done using Ordinary one-way ANOVA using Tukey's multiple comparisons tests (*$p = 0.0439$). (D) Mitotic stability of a monocentric plasmid (pRS313) was determined in wild-type, *atg* mutants as indicated, as well as in the kinetochore mutant strain *ctf19Δ*. The assay was done with three independent sets of transformants. Statistical analysis was done using one-way ANOVA using Dunnett's multiple comparisons tests (****$p < 0.0001$). The underlying data for panels C and D can be found in S1 Data.

during starvation [62], neither displayed any TBZ sensitivity nor any reduction in plasmid stability (Figs 1D and S1A). Using GFP-Atg8 processing assay [65], as previously reported, the *atg1Δ* cells were defective for autophagy (absence of free GFP), while both *atg11Δ* and *atg17Δ* cells have reduced levels of autophagy relative to the wild-type (S1B Fig). This suggests that while Atg1 is indispensable for autophagy, it does not play any role in chromosome segregation as shown by our assays. Taken together, we conclude that the loss of Atg11 in *S. cerevisiae* reduces the fidelity of chromosome segregation leading to an increased frequency of chromosome loss.

## Atg11 displays a dynamic association with the SPBs

To understand how Atg11 contributes to MT-dependent processes, we studied the dynamics of Atg11 during the cell cycle progression. We first confirmed the canonical localization of sfGFP-Atg11 onto the vacuolar membrane as punctate signals at all stages of the cell cycle

(S2A Fig). In addition, we observed that Atg11 puncta were also at the interface of vacuole and nucleus during the cell cycle (S2B Fig). A protein-protein interaction network analysis of components involved in autophagy and chromosome segregation identified Spc72, one of the components of SPBs, as a key interactor of Atg11 (S3A Fig) [66], raising the possibility that Atg11 could be associated with SPBs during the cell cycle. Live-cell imaging revealed that in 18 out of 24 cases, Atg11 is transiently localized proximal to the dSPB prior to anaphase onset (Fig 2A). Proximal localization was studied by partial overlap between Atg11 (cyan) and Spc42 (magenta), giving a white-colored overlap between the two proteins.

To strengthen our observations of the transient localization of Atg11 to SPBs, we employed single-molecule tracking (SMT) to study the dynamic binding of Atg11 with SPBs and the cytoplasmic Atg11 puncta. For the SMT analysis, we endogenously tagged *ATG11* with a HaloTag at its C-terminus for the controlled labeling using HaloTag Ligand (JF646-HTL) (Fig 2B) [67]. The time-lapse movies acquired with slow (200 ms time interval) and fast (15 ms time interval) imaging enabled the quantification of survival time and diffusion coefficient, respectively. We quantified the dwell time of Atg11 over SPBs (Spc42-mCherry) and Atg11 puncta (sfGFP-Atg11) in the cytoplasm from 200 ms time interval movies. The cumulative survival time distribution fits well with the double exponential fit (Fig 2C), suggesting two molecular populations: (a) Atg11 molecules bound with long residence time (specific binding) and (b) bound with short residence time (non-specific binding). We observed specific binding of Atg11 at the cytoplasmic puncta and SPBs ($12.3 \pm 1.1\%$ and $14.5 \pm 2\%$, respectively, Fig 2C). The mean residence time of Atg11 at SPBs was half as compared to the cytoplasmic puncta ($4.4 \pm 0.47$ s versus $8.6 \pm 1.02$ s, Fig 2C), suggesting a fast turnover of Atg11 at SPBs compared to the cytoplasmic puncta. We employed Spot-On-based kinetic modeling [68] for robust estimation of the fraction of bound and unbound molecules with their mean diffusion coefficients (Fig 2D–2F). A marginal increase in the bound fraction of Atg11 over SPBs, compared to the cytoplasm (51% versus 46%, respectively) (Fig 2F) was observed. However, the diffusion coefficient of Atg11 over SPBs is three times higher than that of cytoplasm ($D = 0.013$ µm²/s versus $0.004$ µm²/s) (Fig 2F), suggesting more dynamic interactions of Atg11 with SPBs. We also carried out SMT assays in TBZ-treated cells to understand Atg11 dynamics in the absence of MTs. Under MT-depolymerizing conditions, the specific bound fraction of Atg11 to Atg11-cytoplasmic puncta increased by over 2.5-fold and to SPBs by 2-fold (Fig 2G). The residence time of Atg11 reduces marginally in the presence of TBZ at the Atg11-cytoplasmic puncta ($8.6 \pm 1.02$ s to $4.5 \pm 2.9$ s, Fig 2G), suggesting that the absence of MTs marginally changed its turnover rate. However, the residence time of Atg11 at SPBs increased from $4.4 \pm 0.5$ s to $41.8 \pm 3.9$ s upon TBZ treatment (Fig 2G). Based on these observations, we conclude that the dynamic interactions of Atg11 at the SPBs are MT-dependent (Fig 2H).

To support that Atg11 is transiently localized at the SPBs, we utilized a proximity-based bimolecular fluorescence complementation (BiFC) assay with Spc72, a known interactor of Atg11 (S3A and S3B Fig) [69]. A fluorescent punctate signal resulting from the reconstitution of the Venus molecule at the vacuolar periphery, marked by FM4-64, suggested an *in vivo* interaction between Atg11 and Spc72 (S3B Fig). Co-localization with Spc42 further demonstrated the *in vivo* interaction between these two proteins at the SPBs (S3C and S3D Fig). The absence of BiFC signals in diploid cells expressing either VN-tagged Spc72 or the VC-tagged Atg11 alone (S3E and S3F Fig), suggested that neither VN nor VC fluoresces on its own. To further probe how Atg11 is associated with SPBs, we first confirmed a previously reported interaction between Atg11 and Spc72 [70] by the yeast two-hybrid (Y2H) assay (S3G and S3H Fig). The Y2H assay with Cnm67, a protein from the outer plaque of the SPB, did not show any interactions with Atg11 (S3I Fig), suggesting the spatial position of Atg11 and Spc72 favors their physical interactions. Taken together, the results from the SMT assays and

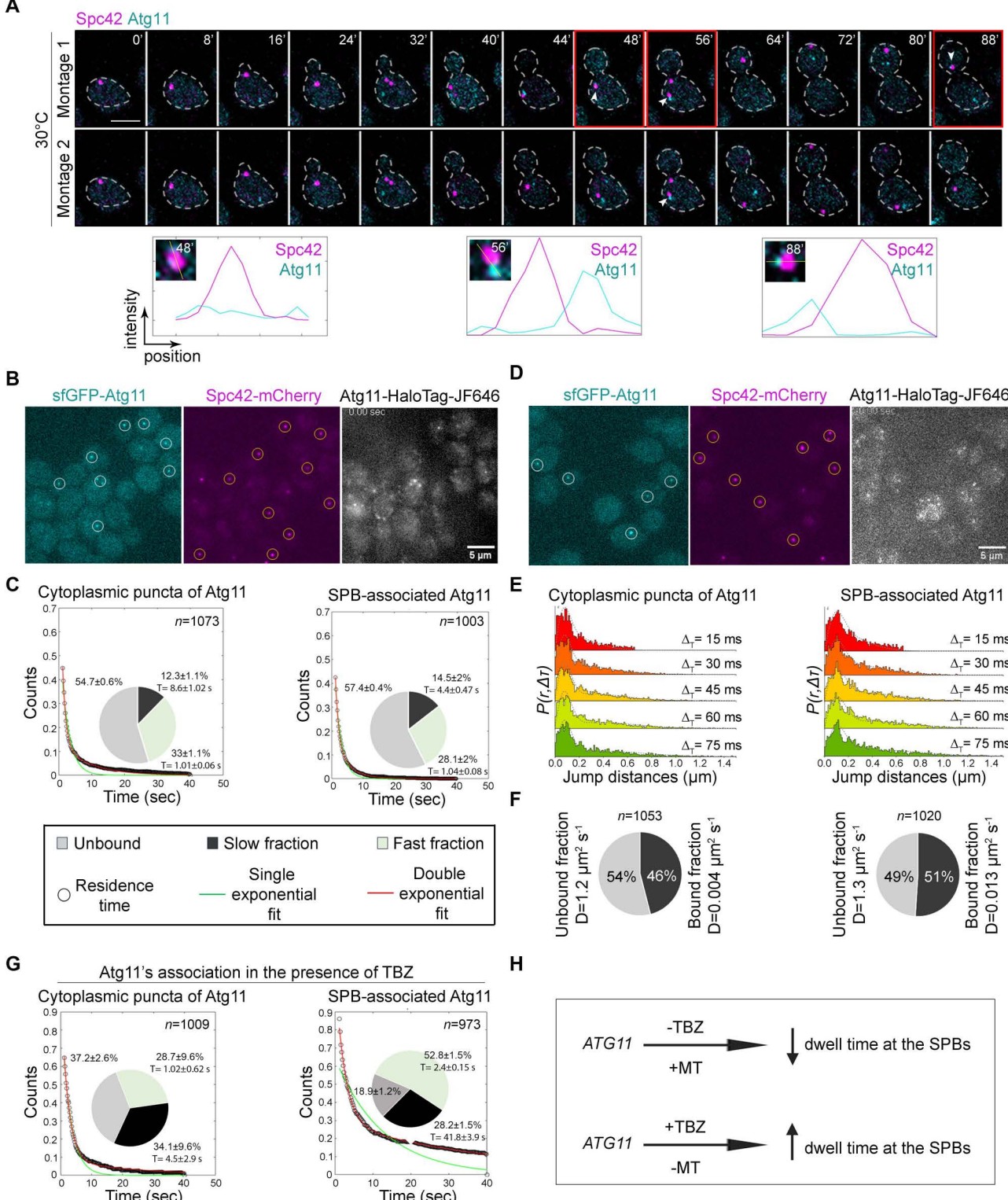

**Fig 2. Single-molecule tracking quantifies the binding dynamics of Atg11 molecules at SPBs. (A)** Time-lapse images showing localization dynamics of Atg11 (sfGFP-Atg11) and SPB (Spc42-mCherry) during the cell cycle after G1 arrest followed by release at 30 °C. Montage 1 and Montage 2 represent two different focal planes, displaying dSPB and mSPB. The white arrowheads denote Atg11's presence proximal to the dSPB. Histogram plots represent the fluorescence intensity profile of sfGFP-Atg11 with Spc42-mCherry at 48, 56, and 88 min (marked with red boundary). Scale bar, 5 μm. **(B)** Representative image for sfGFP-Atg11 puncta in the cytoplasm, Spc42-mCherry, and Atg11-HaloTag-JF646 imaged over 200 ms

intervals. The regions of interest (ROIs) used for tracking Atg11 molecules present in the cytoplasm (cytoplasmic Atg11) or associated with SPBs (SPB-associated) are shown in white and orange circles, respectively. **(C)** Survival time distribution of Atg11 in the cytoplasm and SPB-associated was quantified from 200 ms time-interval movies. The distribution fits well with the double exponential curve, suggesting two types of bound population: (1) fast fraction and (2) slow fraction. Pie charts represent the percentage of molecules unbound (gray), bound with short residence time (fast fraction, light green), and bound with long residence time (slow fraction, black). The average residence times of fast and slow fractions are presented next to their representative fractions. $n$ = number of tracks analyzed. **(D)** Representative image for sfGFP-Atg11 puncta in the cytoplasm, Spc42-mCherry, and Atg11-HaloTag-JF646 imaged over 15 ms intervals. The regions of interest (ROIs) used for tracking Atg11 over cytoplasmic Atg11 and SPBs are shown in white and orange circles, respectively. **(E)** Spot-On based kinetic modeling for Atg11-HaloTag (obtained from 15 ms time interval movies). The probability density function histogram of single-molecule displacements for Atg11 in cytoplasm and over the SPBs is shown. The dashed line indicates the model derived from the cumulative distribution function (CDF) fitting in Spot-On. **(F)** Pie charts represent the fraction of bound and unbound molecules with their mean diffusion coefficients (D), obtained from the Spot-On analysis. $n$ = number of tracks analyzed. **(G)** Survival time distribution of Atg11 at cytoplasmic Atg11 puncta and SPBs was quantified from 200 ms time-interval movies in the presence of TBZ. The distribution fits well with the double exponential curve, suggesting two types of bound population for both: (1) fast fraction and (2) slow fraction. Pie charts represent the percentage of molecules unbound (gray), bound with short residence time (fast fraction, light green), and bound with long residence time (slow fraction, black). The average residence time of fast and slow fractions is presented next to their representative fractions. $n$ = number of tracks analyzed. **(H)** Schematic showing the consequence of TBZ treatment on the dynamics of Atg11 at SPBs using SMT assays. The underlying data for panels C, E, and G can be found in S1 Data.

sub-cellular localization studies, we convincingly demonstrate the existence of a novel and dynamic association of Atg11 at SPBs, in addition to its canonical localization onto the vacuolar membrane crucial for selective autophagy [59].

## Atg11 enables the timely progression of the cell cycle

SPBs nucleate both nuclear and aMTs, critical for kinetochore-MT interactions and spindle orientation along the mother-bud axis, respectively. Having established a non-canonical localization of Atg11 at the SPBs, we, sought to examine the status of both nuclear and aMT regulation in the absence of Atg11. First, we probed cell cycle progression in synchronized cells and studied the dynamics of MTs (GFP-Tub1) and SPBs (Spc42-mCherry) in live cells. We observed a delay in spindle elongation and a concomitant delay in the cell cycle progression in *atg11Δ* cells (Figs 3A, 3B, S4A, and S4B).

We also measured the time taken by *atg11Δ* cells to enter a) metaphase (spindle length = 1.5–2 μm) after release from G1 arrest and b) telophase upon anaphase onset (spindle length > 2 μm). We observed a significant delay at both these stages in *atg11Δ* cells (Figs 3B and S4B). Further analysis revealed that the overnight grown ($t$ = 0) *atg11Δ* cells displayed a significantly increased proportion of large-budded cells compared to the wild-type (S4C Fig). When grown at 37 °C for 6 h, the *atg11Δ* cells further accumulated an increased proportion of large-budded cells (S4C Fig), with a 2-fold higher proportion of large-budded cells with metaphase spindle of 1.5–2 μm compared to the wild-type at 37 °C (Fig 3C), hinting towards a delay in anaphase onset.

Defects in kinetochore-MT attachments lead to the activation of the SAC, an error correction mechanism that delays the metaphase-to-anaphase transition [71]. We, therefore, examined whether the SAC sensed *atg11Δ*-associated defects of delayed anaphase onset. Deletion of *MAD2* in *atg11Δ* cells alleviated the proportion of large-budded cells with unsegregated nuclei demonstrating that indeed SAC senses *atg11Δ*-associated defects leading to delayed anaphase onset (Fig 4A). Cells of *atg11Δ mad2Δ* double mutant grew slower at 37 °C and in the presence of TBZ (Fig 4B). The SAC activation leads to the inhibition of the APC/C-Cdc20 complex, stabilizing securin (Pds1), thus preventing the activation of separase. Inactivation of separase impedes sister chromatid separation and therefore blocks metaphase-to-anaphase transition [72]. Consistent with our observation of SAC-mediated defects, *atg11Δ* cells displayed stabilized Pds1 levels both upon release from nocodazole treatment and G1 arrest (Figs

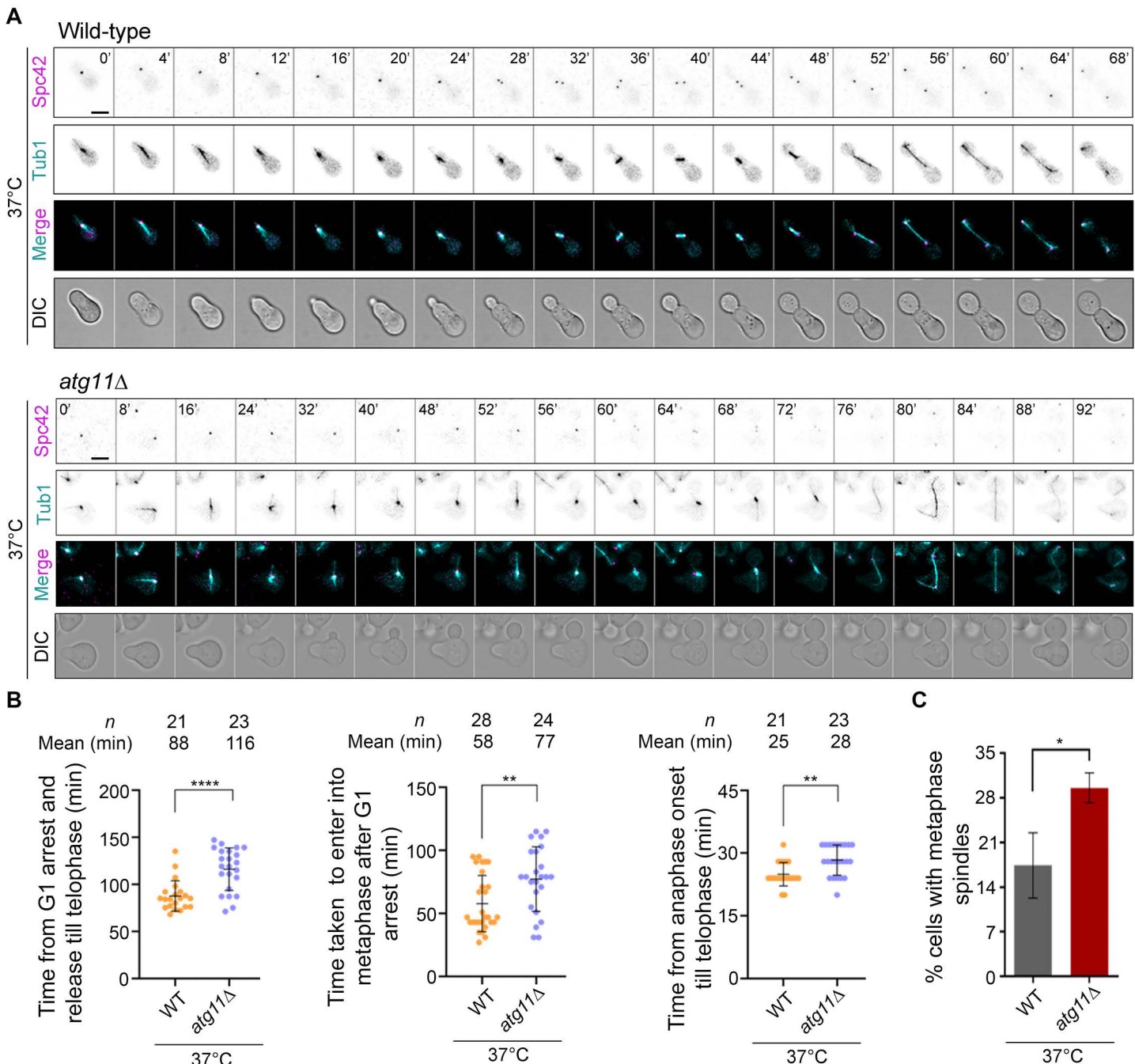

**Fig 3. Cell cycle progression is significantly delayed in *atg11* Δ cells. (A)** Time-lapse images showing dynamics of the mitotic spindle (GFP-Tub1) and SPBs (Spc42-mCherry) during the cell cycle in wild-type (*top*) and *atg11*Δ cells (*bottom*) after G1 arrest followed by release at 37 °C. Scale bar, 5 µm. **(B)** Scatter plot displaying time taken for wild-type and *atg11*Δ cells, after G1 arrest followed by release at 37 °C, for completion of the cell cycle (*left*), to enter into metaphase (*middle*, 1.5–2 µm mitotic spindle), and anaphase onset till telophase (*right*, disassembly of MTs). Error bars show mean ± SD. Statistical analysis was done using an unpaired *t* test with Welch's correction (**$p = 0.0057/0.0011$, ****$p < 0.001$). **(C)** A bar graph representing the proportion of large-budded cells having metaphase (1.5–2 µm) spindle (GFP-Tub1) in wild-type and *atg11*Δ cells at 37 °C. $n > 89$, $N = 3$. Error bars show mean ± SD. Statistical analysis was done using an unpaired *t* test with Welch's correction (*$p = 0.0376$). The underlying data for panels B and C can be found in S1 Data.

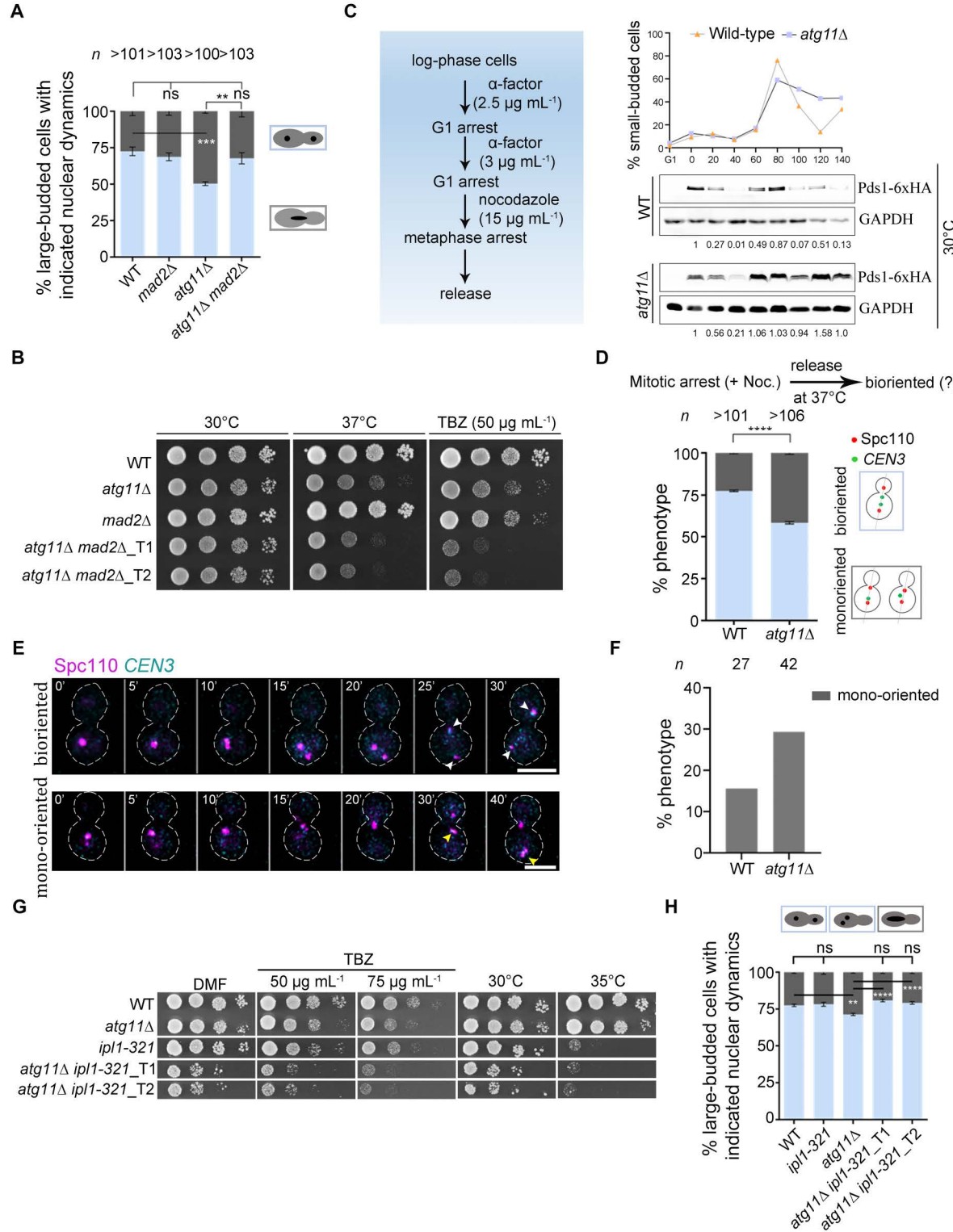

**Fig 4. Ipl1-dependent activation of spindle assembly checkpoint delays cell cycle progression in *atg11* Δ cells. (A)** A bar diagram representing the proportion of large-budded cells with stages of nuclear division in indicated strains when grown at 37 °C for 6 h. Error bars show mean ± SEM. Cartoons represent nuclear morphology (black) in large-budded cells (gray). *n*, a minimum number of cells analyzed, *N* = 4. Statistical analysis was done by two-way ANOVA using Tukey's multiple comparisons test (***p* = 0.0056, ****p* = 0.0006). **(B)** Overnight grown cells of wild-type, single, and double mutants were 10-fold serially diluted and spotted on YPD at 30 and 37 °C and in the presence

of TBZ (50 μg mL$^{-1}$) at 30 °C. Plates were photographed after 48 h of incubation. **(C)** Schematic showing steps involved in the arrest and release of wild-type and *atg11Δ* cells to study Pds1 protein dynamics after metaphase arrest. Western blot analysis shows the expression of Pds1-6xHA in wild-type and *atg11Δ* cells released in YPD at 30 °C after G1 and metaphase arrest using α-factor and nocodazole, respectively. Cells were collected every 20 min to prepare protein samples and to quantify the budding index (*top*). Protein levels of GAPDH were used as a loading control. Pds1 normalized values are indicated below each lane. The experiments were repeated twice with similar results. **(D)** A bar diagram representing the proportion of cells with bioriented (light blue) or mono-oriented kinetochores (*CEN3*-GFP) (dark gray) in each strain grown at 37 °C. *n* represents the minimum number of large-budded cells (budding index of > 0.6) analyzed in three independent biological replicates. Wild-type (*n* > 101), *atg11Δ* (n > 106). Error bars show mean ± SEM. Statistical analysis was done using two-way ANOVA for multiple comparisons (****$p < 0.0001$). **(E)** Time-lapse images of bioriented (*top*, white arrowheads) or mono-oriented (*bottom*, yellow arrowheads) kinetochores (*CEN3*-GFP) and SPBs (Spc110-mCherry) during the cell cycle in wild-type and *atg11Δ* cells after nocodazole arrest followed by release at room temperature. Scale bar, 5 μm. **(F)** A bar diagram representing the proportion of cells with mono-oriented kinetochores (*CEN3*-GFP) (dark gray) obtained from live-cell movies. **(G)** Overnight grown cells of wild-type, single, and double mutants were 10-fold serially diluted and spotted on YPD at 30 and 35 °C and in the presence of TBZ (50 and 75 μg mL$^{-1}$) at 30 °C. Plates were photographed after 33 h of incubation. **(H)** A bar diagram representing the proportion of large-budded cells with stages of nuclear division in indicated strains when grown at 35 °C for 6 h. Error bars show mean ± SEM. Cartoons represent nuclear morphology (black) in large-budded cells (gray). Wild-type (*n* > 101), *atg11Δ* (*n* > 104), *ipl1-321* (*n* > 69), *atg11Δ ipl1-321*_T1 (*n* > 101), *atg11Δ ipl1-321*_T2 (*n* > 102). *n*, a minimum number of cells analyzed, *N* = 3. Statistical analysis was done by two-way ANOVA using Tukey's multiple comparisons test (**$p = 0.0055$, ****$p < 0.0001$). The underlying data for panels A, C–D, F, and H can be found in S1 Data. Uncropped western blots are available in S1 Raw Images.

4C and S5A). Our results, therefore, provide evidence that *atg11Δ* cells fail to establish correct kinetochore-MT attachments leading to a delayed anaphase onset.

Does Atg11 help in maintaining the structural integrity of kinetochores or in regulating MT dynamics? The *CEN* transcriptional read-through assay [73] ruled out the role of Atg11 in the maintenance of the kinetochore ensemble (S5B Fig). The kinetochore and SAC mutants display growth defects in the presence of MT poisons due to a defect in chromosome segregation. An increased chromosome mis-segregation upon loss of Atg11 and the metaphase arrest of *atg11Δ* cells in the presence of TBZ suggest that the fidelity of chromosome segregation is compromised. To test this, we monitored the segregation dynamics of a marked chromosome in a strain with *CEN3* labeled with LacO repeats and LacI tagged with GFP [74]. Most wild-type cells displayed bioriented sister kinetochores, as studied by co-localizing Spc110-mCherry with *CEN3*-GFP, after recovery following nocodazole treatment (Figs 4D and S5C). However, a significant increase in mono-oriented kinetochores was observed in *atg11Δ* cells. This was further corroborated by live-cell imaging after nocodazole arrest and release (Fig 4E and 4F). Mono-oriented kinetochores do not generate tension and are sensed by Aurora kinase B$^{Ipl1}$, which activates SAC by creating unattached kinetochores [10]. With an increased proportion of cells displaying mono-oriented kinetochores and to determine whether the SAC-dependent delay in the cell cycle in *atg11Δ* cells is due to Ipl1 activation, we deleted *ATG11* in *ipl1-321* mutant cells [75]. Cells of *atg11Δ ipl1-321* double mutant grew slower both at 30 °C and in the presence of TBZ (Fig 4G). These cells failed to arrest and exhibited nuclear mis-segregation defects (Fig 4H), suggesting that the SAC-dependent delay in *atg11Δ* is due to Ipl1-mediated improper kinetochore–MT interactions. We also asked if Atg11 mediates this function by localizing to the kinetochores by SMT assays. We carried out SMT assays in a strain co-expressing *ATG11*-Halo and *MTW1*-mCherry. Our results show no specific binding of Atg11 at the kinetochores (S5D Fig), suggesting that Atg11 primarily functions at SPBs in mediating MT dynamics during the cell cycle.

Fluorescence recovery after photobleaching (FRAP) of the metaphase spindle established that the kinetochore MTs are dynamic in budding yeast [76]. We examined if there is any perturbation in the dynamics of the metaphase spindle upon loss of *ATG11*. The wild-type cells exhibited 50% fluorescence recovery of the bleached region after 300 s (*n* = 12 live-cells) (Fig 5A), while the *atg11Δ* cells took 338 s (*n* = 11 live-cells) (Fig 5B). The corrected ratio of FRAP between the bleached and unbleached regions of the spindle after full recovery was

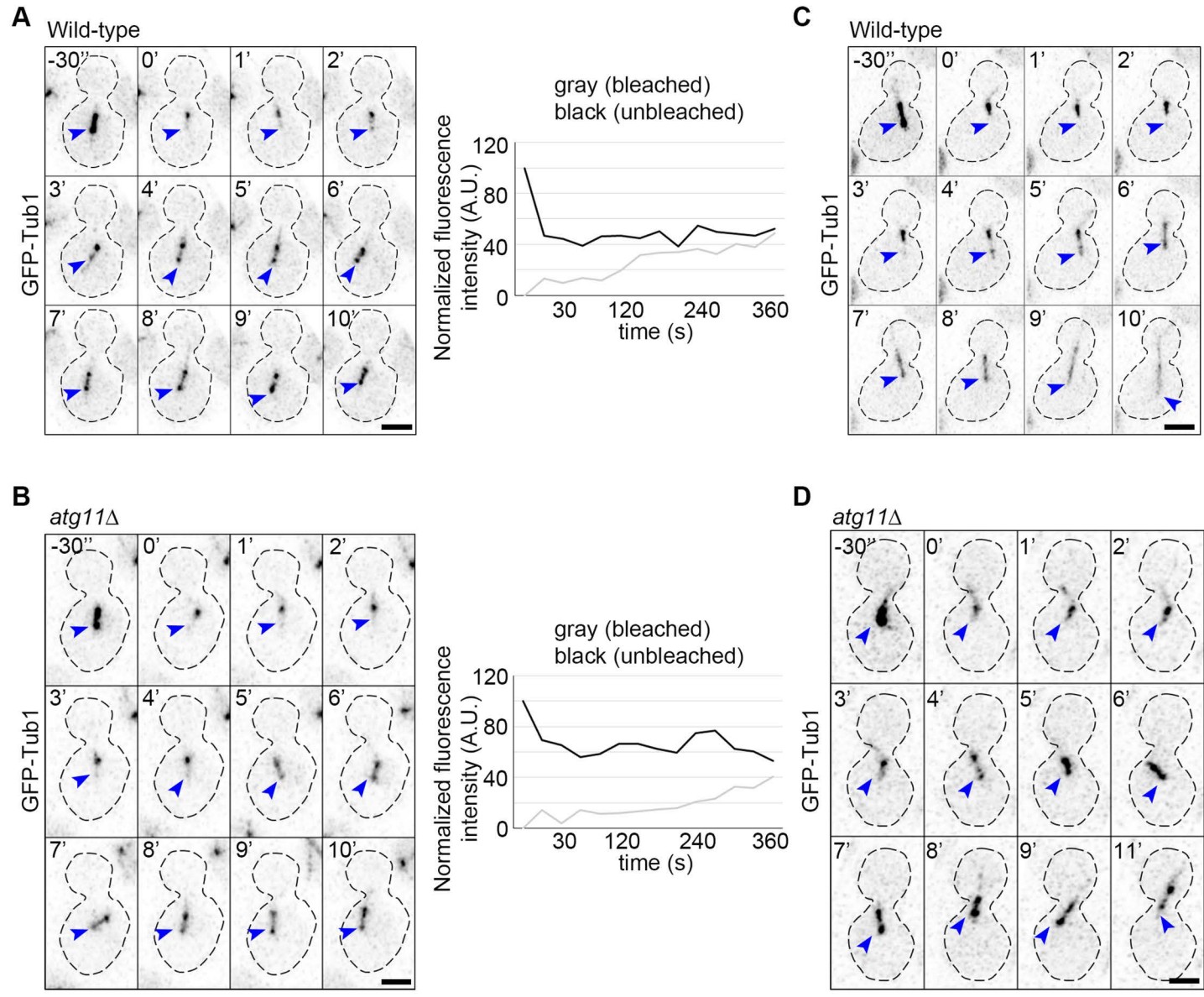

**Fig 5. Fluorescence recovery after photobleaching (FRAP) of the metaphase spindle. (A, B)** Time-lapse images (*left*) and graphs (*right*) representing the fluorescence recovery of photobleached metaphase spindles (blue arrowheads) in (A) wild-type and (B) *atg11Δ* cells. **(C, D)** Time-lapse images depicting the fluorescence recovery of metaphase spindle (blue arrowheads) after photobleaching and progression to anaphase in wild-type (C), and migration to the daughter bud in *atg11Δ* cells (D). $t =$ −30 s shows the metaphase spindle before photobleaching, whereas the $t = 0$ time point represents the bleached lower-half spindle. Scale bar, 2 μm. The graphs represent the quantification of fluorescence intensities of both halves (as shown) of the metaphase spindle with time. The underlying data for panels A and B can be found in S1 Data.

approximately 70% for both wild-type and *atg11Δ* cells. As shown previously, the photo-bleached cells proceeded through anaphase, suggesting that bleaching of the metaphase spindle did not damage the spindle (Fig 5C). However, we observed the migration of the metaphase spindle to daughter cells after recovery only in *atg11Δ* cells (Fig 5D). Overall, we observed a marginal delay of 38 s in *atg11Δ* cells to achieve 50% recovery of the bleached region. In summary, our results reveal that Atg11 plays a critical role in cell cycle progression

in conditions that destabilize MTs either by nocodazole treatment or by growing cells at 37 °C. The frequency of erroneous kinetochore-MT attachments increases in the absence of Atg11 under such conditions.

As cells progress from metaphase to anaphase, the serine/threonine phosphatase, Cdc14, is first released from the nucleolus into the nucleus by the Cdc Fourteen Early Anaphase Release (FEAR) network in yeast [77, 78]. Cdc14 release is required to inactivate CDKs during exit from mitosis [79–82]. While securin (Pds1) and nucleolar protein Fob1 negatively regulate the FEAR network, the same network is positively regulated by separase Esp1, the kinetochore/spindle protein Slk19, Spo12, Bns1, and the polo kinase Cdc5 [80,83–86]. Since Pds1 levels are found to be stabilized in the absence of Atg11, we examined whether the *atg11Δ* cells display any delay in Cdc14 release. We probed for Cdc14 (Cdc-14-GFP) dynamics in alpha-factor synchronized G1 cells. Indeed, *atg11Δ* cells showed a significant delay in Cdc14 release from the nucleolus (S6A and S6B Fig).

The FEAR pathway regulates multiple anaphase events, including timely exit from mitosis, midzone spindle stabilization, segregation of ribosomal DNA, and nuclear positioning [87]. Separase and Slk19 regulate the stabilization of fragile spindle midzone during anaphase by targeting the Ipl1-Sli15-Bir1 complex to the spindle midzone in early anaphase, which in turn facilitates the recruitment of the spindle-stabilizing protein Slk19 [88–90]. The absence of either separase or Slk19 leads to the collapse of anaphase spindle. We observed MT buckling in *atg11Δ* cells at anaphase, although in a small proportion of cells (5% of the live-cell) (Fig 6A). The FEAR network is also vital for nuclear positioning and is required to activate pulling forces at aMTs in the mother cell during anaphase. Loss-of-function of separase results in the movement of mitotic spindles into daughter cells called the 'daughterly phenotype' [91]. *atg11Δ* cells displayed this phenotype in 14% of the live cells analyzed ($n = 29$ live-cell) (Fig 6B), along with anaphase spindle elongation in the mother cell (Fig 6B, 10% of the live cells). Kar9, a MAP, that regulates nuclear positioning at metaphase, is known to be dephosphorylated in early anaphase and is suggested to be a target of Cdc14 [28, 29]. We, therefore, examined whether Atg11 influences the Kar9- or dynein-dependent pathway of spindle positioning by examining the effect of low temperature, known to enhance the binucleated cell phenotype [43]. The *dyn1Δ*, *kip2Δ*, and *dyn1Δ kip2Δ* double mutant strains exhibited a significant increase in the proportion of binucleated cells as expected (Fig 6C) [92]. Remarkably, *atg11Δ dyn1Δ* double mutant cells had a significantly increased proportion of binucleated cells. These cells also displayed mitotic exit with mispositioned spindles, marked by the accumulation of multinucleated and anucleate cells (Fig 6C). The *kar9Δ* and *atg11Δ kar9Δ* mutants had more binucleated cells than those of the wild-type (Fig 6C). However, the proportion of binucleated cells of the *atg11Δ kar9* double mutant was similar to *kar9Δ* (Fig 6C), suggesting that Atg11 contributes to spindle positioning in a Kar9-dependent manner. Since Kar9 regulates spindle positioning at metaphase, we, therefore, examined whether the inactivation of Atg11 perturbed the spindle alignment and migration of metaphase spindles to the bud-neck. We measured the distance between the metaphase spindle and the bud-neck along with the angle between the spindle axis and the mother-bud axis. The *atg11Δ* cells displayed a 2-fold increase in the proportion of cells having a metaphase spindle at the central region in the mother cell (Fig 6D), as well as a significantly higher angle of alignment of the metaphase spindle than the wild-type (Fig 6E).

Kar9 is asymmetrically localized at the dSPB, which in turn is required for correct spindle positioning in the wild-type cells. Therefore, we examined if the defects in spindle positioning upon loss of Atg11 are due to alterations in the asymmetric localization of Kar9 at the dSPB. To address this, we analyzed the distribution of Kar9, tagged with GFP, in metaphase cells co-expressing Spc72-mCherry that marks the SPB. While 16% of wild-type cells displayed

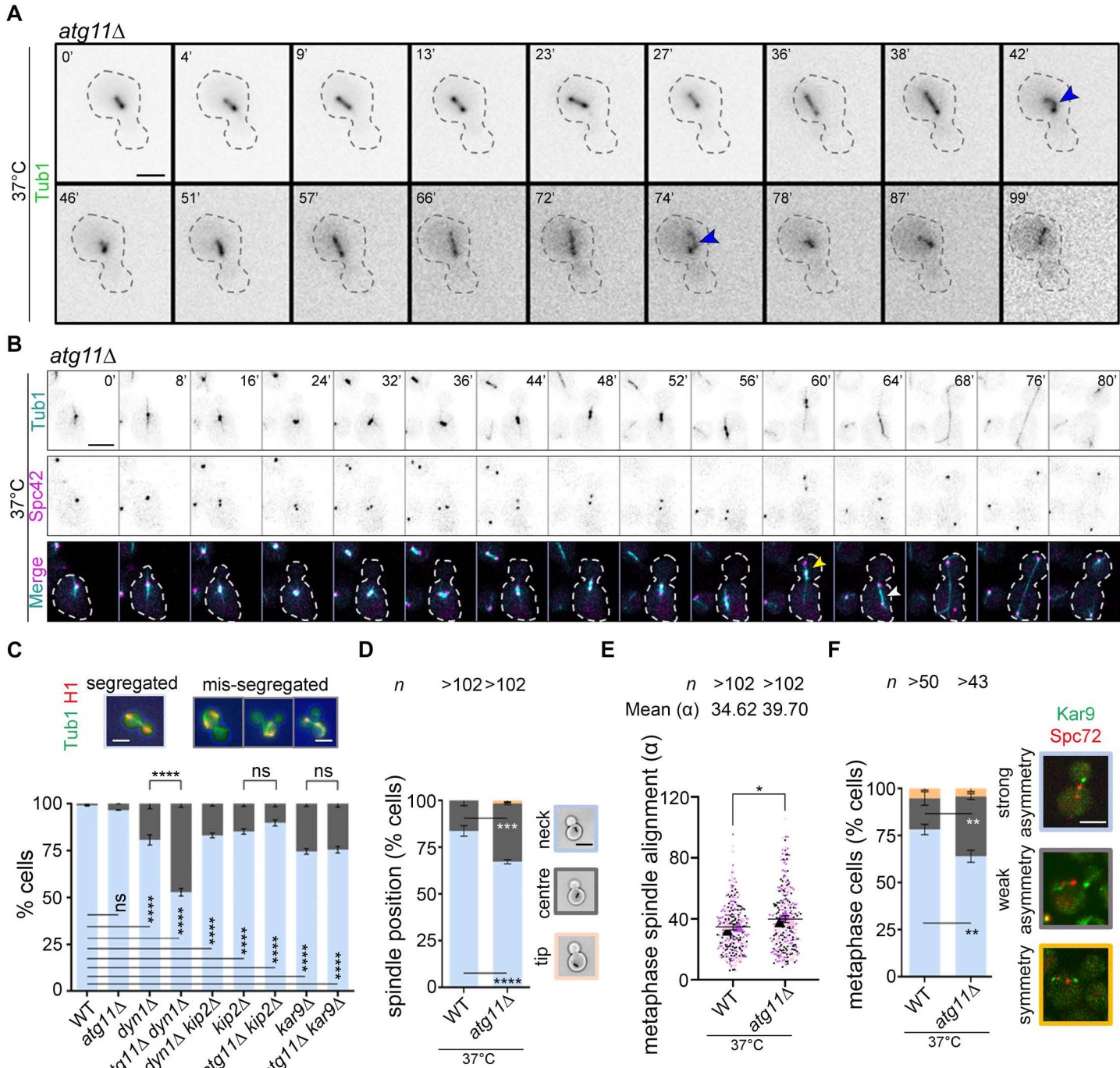

**Fig 6. The *atg11* Δ cells exhibit Kar9-dependent spindle positioning and alignment defects. (A)** Time-lapse images showing spindle (GFP-Tub1) buckling defects (blue arrowheads) at 37 °C. **(B)** Time-lapse images showing the mitotic spindle (GFP-Tub1) and SPB (Spc42-mCherry) dynamics during the cell cycle in *atg11*Δ cells after G1 arrest followed by release at 37 °C. Yellow and white arrowheads mark the movement of the mitotic spindle completely into the daughter cell and the complete elongation of the anaphase spindle in the mother cell, respectively. Scale bar, 5 μm. **(C)** A bar diagram showing the proportion of cells with properly segregated nuclei (light blue) or with improperly segregated nuclei represented by binucleated or multinucleated cells (dark gray). Error bars show mean ± standard error of mean (SEM). Statistical analysis was done using two-way ANOVA using Tukey's multiple comparisons test (****$p < 0.0001$). Scale bar, 5 μm. **(D)** The position of the short bipolar spindle in metaphase cells of wild-type (WT) and *atg11*Δ grown at 37 °C were analyzed. *n*, a minimum number of cells analyzed, $N = 3$. Statistical analysis was done by two-way ANOVA using Sidak's multiple comparison test (***$p = 0.0001$ and ****$p < 0.0001$). Scale bar, 5 μm. **(E)** The angle of alignment along the mother–bud axis in metaphase cells of wild-type (WT) and *atg11*Δ grown at 37 °C and displaying a short bipolar spindle were analyzed. *n*, a minimum number of cells analyzed, $N = 3$. The statistical significance was done using an unpaired *t* test with Welch's correction (*$p = 0.0446$). **(F)** A bar diagram showing the proportion of metaphase cells carrying

Kar9-GFP either on one SPB (strong asymmetry), both the SPBs unequally (weak asymmetry), or both the SPBs equally (symmetry) in the indicated strains. Metaphase cells were identified by measuring the distance between the two SPBs (marked by Spc72-mCherry puncta). Statistical analysis was done using two-way ANOVA using Sidak's multiple comparisons test (**$p$ = 0.0030/0.0050). Scale bar, 5 μm. The underlying data for panels C–F can be found in S1 Data.

weak asymmetric Kar9 localization, *atg11Δ* cells displayed a 2-fold increase in the proportion of cells with weak asymmetric Kar9 localization (Fig 6F). This loss in Kar9 asymmetry in *atg11Δ* cells explains the 2-fold increase in the proportion of cells having a metaphase spindle at the central region in the mother cell. Taking these observations together, our results suggest that Atg11 facilitates timely Cdc14 release from the nucleolus during early anaphase upon timely degradation of securin, which is necessary for spindle stabilization and Kar9-dependent spindle positioning and alignment in budding yeast.

## Atg11 facilitates dynamic instability of aMTs via timely degradation of Clb4

Dynamic instability of MTs also facilitates the interaction of MTs with the bud cortex, crucial for spindle orientation and positioning in budding yeast. Various factors like motor proteins, MT-binding proteins, and MT-nucleation proteins regulate MT dynamics and proper positioning of the mitotic spindle [92–94]. We, therefore, sought to understand if Atg11 additionally contributes to the dynamic instability of aMTs in budding yeast by being localized at the SPBs. We studied aMT (GFP-Tub1) dynamics in metaphase cells of wild-type and *atg11Δ* cells (Figs 7A and 7B). The *atg11Δ* cells displayed a significantly increased catastrophe frequency (Fig 7C), while the rescue frequencies were comparable to those of the wild-type (Fig 7C). A closer examination revealed a significant increase in the number of catastrophe events, decreased aMT length at catastrophe and lower time spent in polymerization before catastrophe in *atg11Δ* cells (S7A, S7B and S7C Fig). The *atg11Δ* cells also showed a significant increase in the number of rescue events (S7D Fig). However, there was no difference observed in the minimum length of aMTs and the time spent for depolymerization before the rescue onset between wild-type and *atg11Δ* cells (S7E and S7F Fig). Remarkably, loss of Atg11 lowered both polymerization and depolymerization rates of aMTs in *atg11Δ* cells (Fig 7C). Consequently, the length of aMTs in *atg11Δ* cells was also significantly shorter at the metaphase and anaphase stages of the cell cycle (S7G–S7I Fig). These results revealed that Atg11 facilitates dynamic instability of aMTs and prevents catastrophe frequency in budding yeast.

The aMT dynamics are tightly regulated in *S. cerevisiae*, switching from an unstable to a stable state at anaphase onset. This transition requires the degradation of the mitotic cyclin Clb4 by the APC/C$^{Cdc20}$ complex via the proteasomal degradation pathway, which is crucial for the establishment of proper spindle positioning [95]. The unstable aMTs in *atg11Δ* cells led us to investigate if the Clb4 levels were altered. The wild-type cells showed cycling levels of Clb4, being highest at metaphase and levels dropped as cells entered anaphase both at 37 and 30 °C (Figs 7D and S8A). Consistent with our observation of unstable aMTs in *atg11Δ* cells, we observed stabilized Clb4 levels with loss of Atg11 (Figs 7D and S8A). Taken together, these results suggest that *atg11Δ* cells fail to facilitate the timely degradation of Clb4 crucial for aMT stability at anaphase onset in budding yeast.

Both asymmetric localization of Kar9 at the dSPB and proper integrity of aMTs ensure the asymmetric inheritance of the old SPB (dSPB) into the daughter bud [28,36]. With a random Kar9 distribution during metaphase and shorter aMTs due to loss of Atg11, we examined whether *atg11Δ* cells display any defects in SPB inheritance. To test this possibility, we tagged Spc42 (SPB marker) with mCherry. The slow folding kinetics of mCherry and ordered

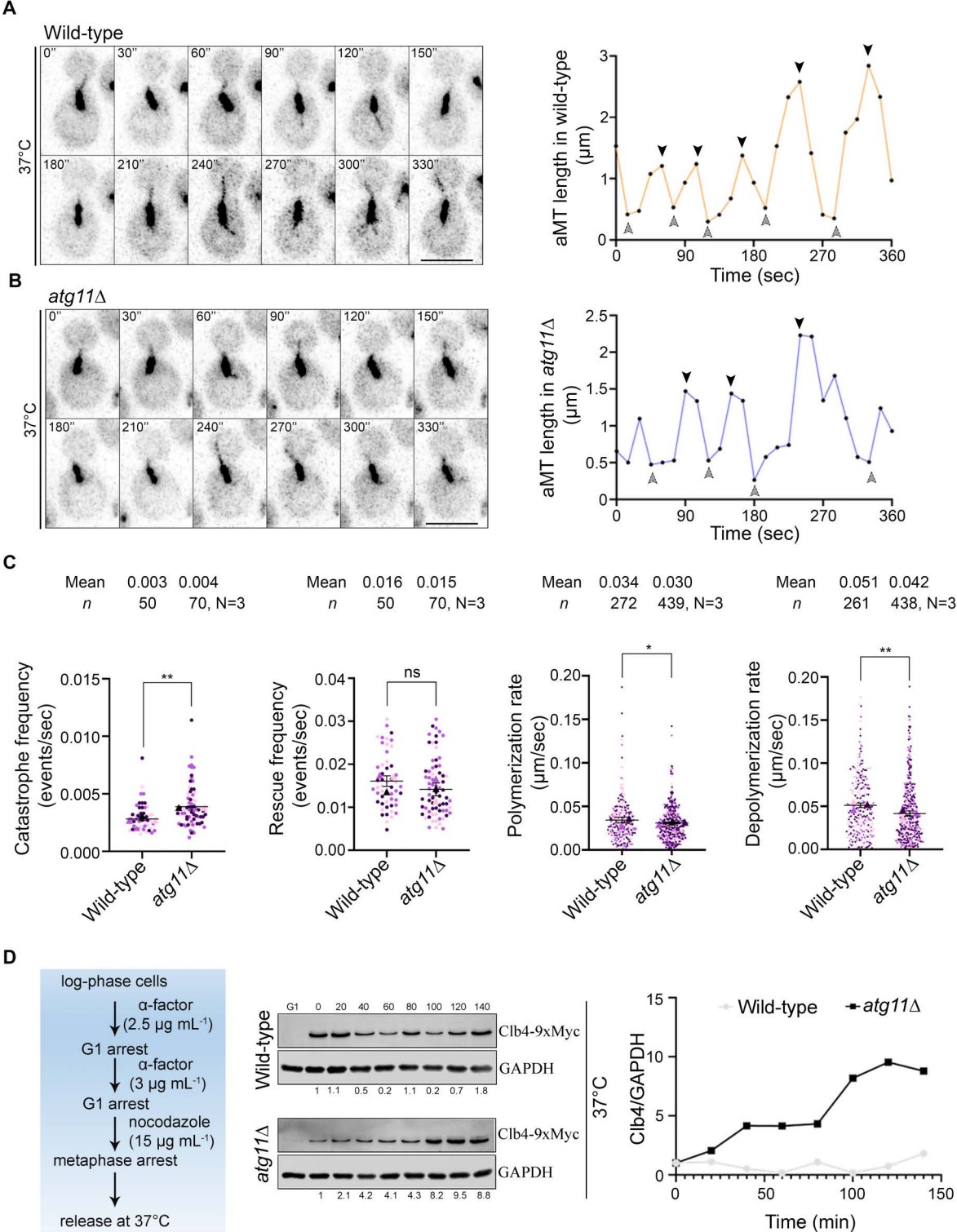

**Fig 7. Cells lacking Atg11 have stabilized Clb4 levels leading to altered astral microtubule dynamics. (A)** Time-lapse images (*left*) and graphs (*right*) showing the length of astral MTs (aMTs) (GFP-Tub1) in pre-anaphase cells of wild-type cells after G1 arrest and release at 37 °C. **(B)** Similar images (*left*) and graphs (*right*) are shown for *atg11Δ* cells grown and analyzed under identical conditions as in the wild-type. Scale bar, 5 μm. Black and gray arrowheads represent the catastrophe and rescue events, respectively. **(C)** Scatter plot displaying catastrophe frequency, rescue frequency, polymerization rate, and depolymerization rate in wild-type and *atg11Δ* cells as indicated.

Error bars show mean ± standard error of mean (SEM). Statistical analysis was done using an unpaired *t* test with Welch's correction (**$p = 0.0022/0.0015$, *$p = 0.0443$). **(D)** Schematic (*left*) showing steps involved in synchronizing followed by the release of wild-type and *atg11Δ* cells to study Clb4 protein dynamics after G1 and metaphase arrest and release at 37 °C. Cells were collected every 20 min to prepare protein samples. Western blot analysis (*right*) shows the expression of Clb4-9xMyc in wild-type and *atg11Δ* cells. Protein levels of GAPDH were used as a loading control. Clb4 normalized values are indicated below each lane and the values are plotted as a line graph. Experiments were repeated twice with similar dynamics. The underlying data for panels A–D can be found in S1 Data. Uncropped western blots are available in S1 Raw Images.

assembly of SPBs allow age discrimination with the new SPB (mSPB) being significantly dimmer than the dSPB [51]. Deletion of *KAR9*, known to display random SPB inheritance [36], led to a randomized SPB inheritance relative to the wild-type (Fig 8A). Strikingly, *atg11Δ* cells showed a 4-fold increase in SPB inheritance defects (Fig 8A). Notably, autophagy mutant cells lacking a key autophagy protein Atg1 segregated SPBs like in the wild-type cells (Fig 8A), hinting that SPB inheritance is autophagy-independent in budding yeast. A previous report on the localization interdependency of Spc72 and Kar9 onto the SPBs, placed Spc72 upstream of Kar9 [96], while a recent study demonstrates Cdc5 as a molecular timer facilitating timely recruitment of both Spc72 and Kar9 onto the SPBs [97]. Based on the SPB inheritance defects, we asked whether Atg11 contributes to the asymmetric localization of Spc72 and Kar9 at the SPBs. Strikingly, the loss of *ATG11* randomized the SPB localization of Spc72 (Fig 8B and 8C). Taken

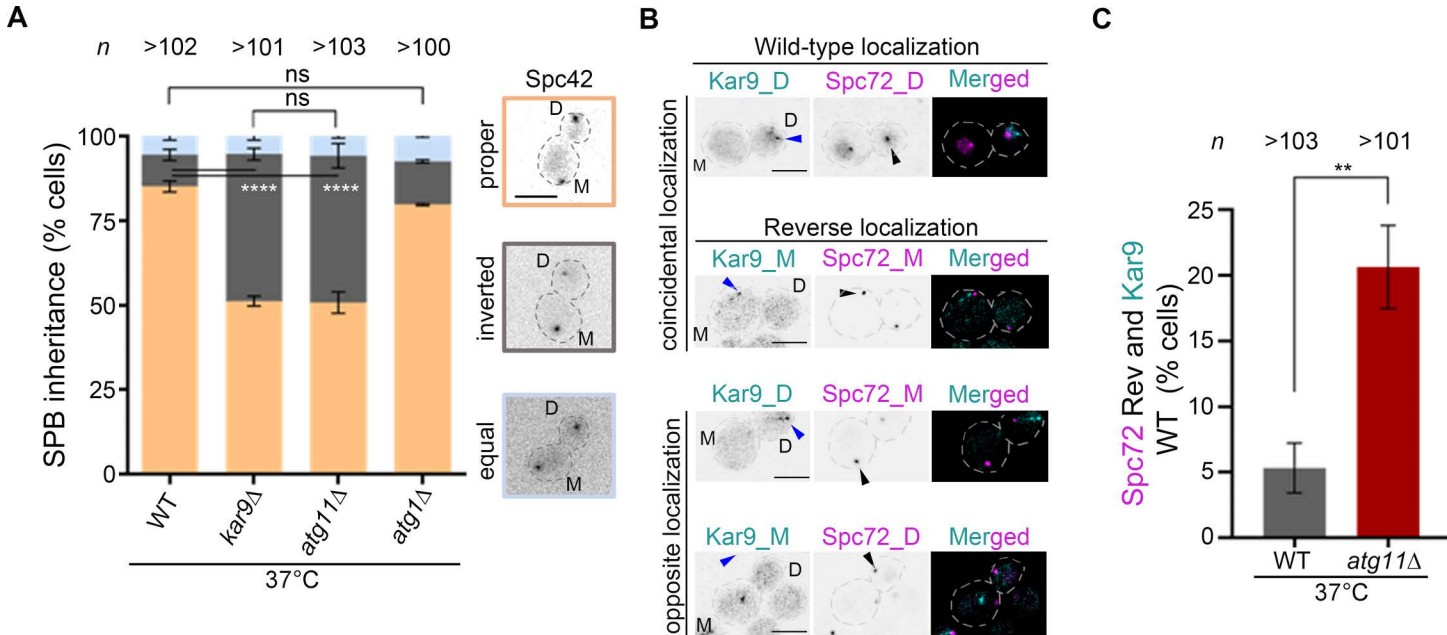

**Fig 8. Spc72-mediated SPB inheritance is significantly altered in *atg11* Δ cells. (A)** A bar diagram displaying the proportion of cells having proper (dSPB in the daughter cell), reversed (dSPB in the mother cell), or symmetric (SPBs are indistinguishable) SPB inheritance in the indicated strains based on the Spc42-mCherry signal intensity at 37 °C. *Right*, representative images of above mentioned SPB inheritance are shown. M and D represent the mother and daughter cells, respectively. *n*, a minimum number of cells analyzed, *N* = 3. Statistical analysis was done by two-way ANOVA using Dunnett's multiple comparisons test (****$p < 0.0001$). Scale bar, 5 μm. **(B)** Representative images displaying coincidental or opposite distribution of Kar9 (cyan) and Spc72 (magenta) in wild-type (WT) and *atg11Δ* cells, exhibiting wild-type-like asymmetric and predominant accumulation of both proteins in the daughter bud (D) or reversed such that asymmetric and predominant accumulation of these proteins in the mother (M) cell. Asymmetric localization of Kar9-GFP and Spc72-mCherry is marked by blue and black arrowheads, respectively. Scale bar, 5 μm. **(C)** Bar diagram representing the proportion of cells displaying inverted Spc72 localization when Kar9 is asymmetrically localized in daughter cells at 37 °C. *n*, a minimum number of cells analyzed, *N* = 3. Error bars show mean ± SD. The statistical significance was done using an unpaired *t* test with Welch's correction (**$p = 0.0041$). The underlying data for panels A and C can be found in S1 Data.

together, we conclude that being localized at the SPBs, Atg11 prevents catastrophe frequency of aMTs, crucial for proper aMT length and asymmetric SPB inheritance in budding yeast.

## Discussion

We unravel an unusual function of an autophagy-related protein, Atg11, in maintaining the high-fidelity transmission of genetic material through many generations. Our results demonstrate that *atg11Δ* cells show a high rate of plasmid and chromosome loss. How does an autophagy-related protein, predominantly distributed in the cytoplasm, regulate chromosome segregation? Strikingly, SMT provides evidence of a significant and dynamic pool of Atg11 molecules associated with SPB which is dependent on the presence of MT. Loss of Atg11 led to the accumulation of a large proportion of cells with mono-oriented chromosomes and Ipl1-dependent SAC-mediated delay in cell cycle progression. This is supported by an increase in mis-segregated nuclei in *atg11Δ ipl1–321* cells and by the stability of securin protein Pds1 observed in *atg11Δ* mutants. *atg11Δ* cells show the appearance of FEAR-defective phenotypes of spindle destabilization and loss of Kar9-dependent nuclear positioning. In addition, Clb4 levels are stabilized and aMT catastrophe is significantly altered in *atg11Δ* cells. This is evidenced by a higher proportion of cells with short aMTs. Without Atg11, many cells lost asymmetric localization of both Spc72 and Kar9, leading to the loss of asymmetric SPB inheritance. We propose that by localizing at SPBs, Atg11 likely mediates the biased distribution of factors between daughter and mother SPBs. This impacts MT dynamics, both in the nucleus and cytoplasm and a set of coordinated events that ensure precise chromosome segregation (Fig 9). The non-canonical role of Atg11 in maintaining ploidy is significant, especially in light of an increasing body of evidence that suggests defects in MT-mediated cellular processes generate aneuploidy, a cellular state commonly observed in diseases such as cancer [101–106].

Based on the SMT assay, our study validates two distinct pools of Atg11 in the budding yeast, including a less dynamic cytoplasmic pool of Atg11 required for selective autophagy at the PAS which is actin-dependent [98,99]. Consisting of four coiled-coil domains, Atg11 functions as a scaffolding protein, with the fourth coiled-coil domain (CC4) of Atg11 important for selective autophagy [59]. The second pool of Atg11, being more dynamic, uncovers a non-canonical localization of Atg11 to SPBs. Remarkably, the dynamics of single molecule pools of Atg11 at the SPBs are MT-dependent (Fig 9), as they display a significant increase in residence time at the SPBs in the absence of MT upon TBZ treatment. The interaction of Atg11 with Spc72 suggests a repurposing of the scaffolding function of Atg11 in budding yeast, which undergoes closed mitosis. We posit that being a bridging protein, Atg11 likely interacts, recruits, or positions proteins key to the asymmetric distribution of Spc72. The polo-like kinase, Cdc5, facilitates the asymmetric localization of Spc72 and Kar9 at the dSPB crucial for asymmetric SPB inheritance [97]. Cdc5 also displays a physical association with Spc72 and safeguards SPB asymmetry [107]. Our study suggests the loss of non-random SPB inheritance in the absence of Atg11. Therefore, it is tempting to speculate that Atg11 and Cdc5 may interact *in vivo* and such an interaction facilitates asymmetric localization of Spc72 and Kar9. It remains to be seen if Atg11 brings Cdc5 to dSPB or vice-versa.

The alignment of the mitotic spindle along the polarity axis ensures proper inheritance of DNA between the two cells during asymmetric cell division. The asymmetric localization of Kar9 at the dSPBs together with long and dynamic aMTs is crucial for the positioning and alignment of the pre-anaphase spindle. Bim1, the yeast homolog of the plus-end tracking protein EB1, mediates the interaction of Kar9 at the SPBs and MTs [28]. Bim1-Kar9 guides the aMTs into the daughter bud via interaction with Myo2, which moves along polarized actin filaments [22]. Consistent with this, a loss of function of proteins affecting either aMT dynamics or positioning of pre-anaphase spindle displays misaligned and mispositioned mitotic spindle

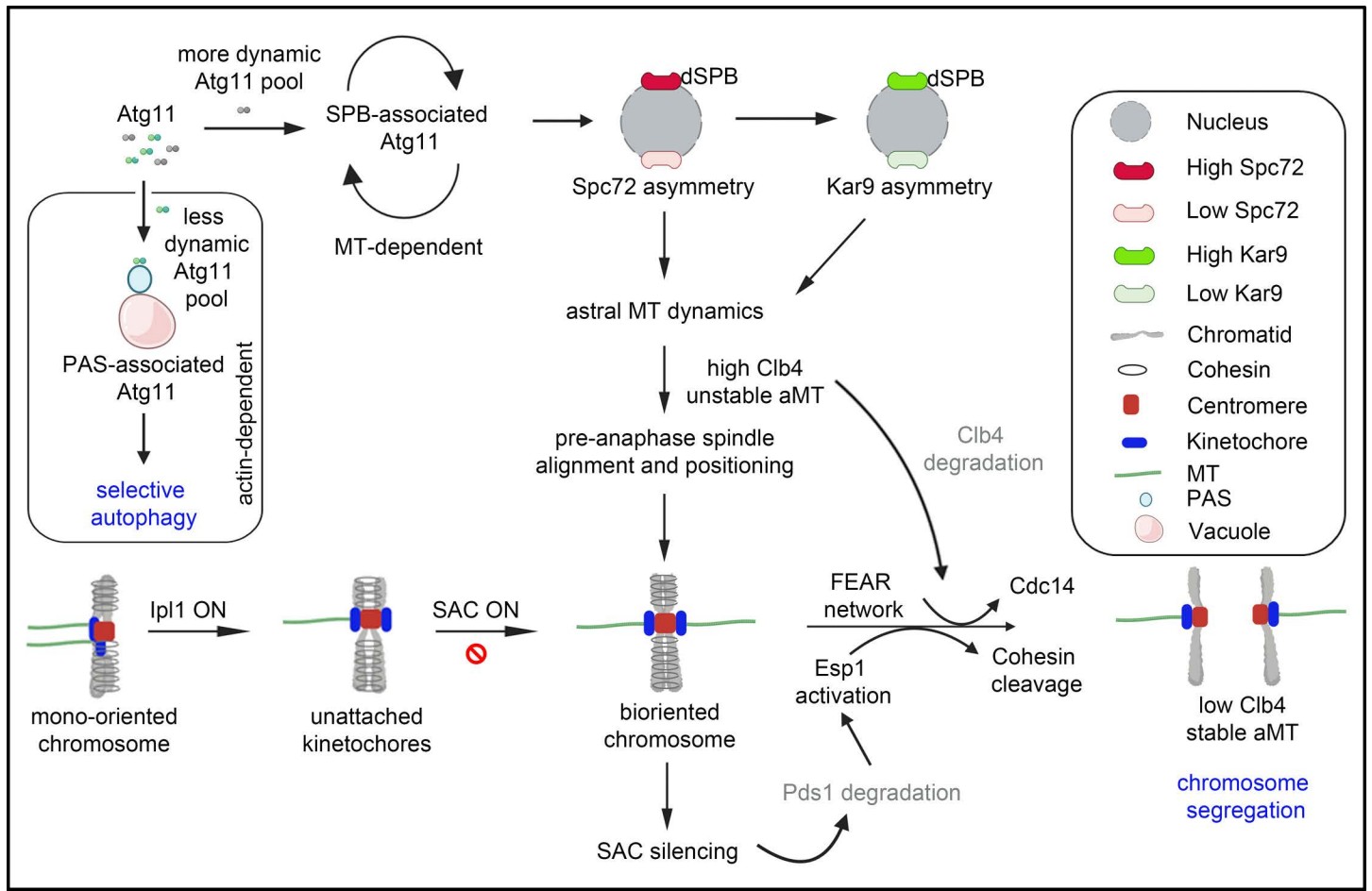

**Fig 9. SPB-associated Atg11 ensures high-fidelity chromosome segregation in budding yeast.** This study reveals two distinct pools of Atg11 in *S. cerevisiae*. A lower proportion of Atg11 which is less dynamic canonically localizes at the pre-autophagosomal structures (PAS) onto the vacuolar membrane [59], necessary for selective autophagy and dependent on actin [98,99]. While the more dynamic Atg11 localizes at the SPBs in a higher proportion which is critical for high-fidelity chromosome segregation. The asymmetric Kar9 localization together with the dynamic instability of MTs is crucial for pre-anaphase spindle positioning and alignment, and chromosome biorientation. Before anaphase onset, aMTs are unstable and cyclin Clb4 levels are high. The mono-oriented kinetochores have defective tension, leading to the activation of Ipl1. Ipl1 carries out the disassembly of defective kinetochore-MT attachments via phosphorylation and activates SAC. SAC delays anaphase onset until biorientation is achieved. Upon biorientation, SAC is silenced, Pds1 gets degraded, separase is activated, and Cdc14 is released from the nucleolus via the FEAR pathway leading to proper sister chromatid separation [100]. APC-C/Cdc20 also degrades Clb4 upon anaphase onset crucial for the stability of aMTs. Atg11 ensures asymmetric localization of both Spc72 and Kar9 at the dSPB, critical for non-random SPB inheritance.

[21,24,37,93,94,108]. Atg11 adds to the list of proteins impacting Kar9 asymmetry and hence MT dynamics in *S. cerevisiae* (Fig 9). The non-random localization of Kar9 at the dSPB is regulated by three independent pathways: Cdk1-dependent phosphorylation, sumoylation, and SAC signaling [33]. Clb4-Cdk1 regulates Kar9 asymmetry by phosphorylating Kar9 at dSPB, and reduces its affinity with Bim1, thereby inhibiting the association of Kar9 with SPBs and MTs [28]. Astral MT dynamics are regulated by two evolutionary conserved machineries, Clb4-Cdk1, and APC/C$^{Cdc20}$ complexes. aMTs are stabilized at the anaphase onset by the APC/C$^{Cdc20}$-mediated degradation of mitotic cyclin Clb4 [95]. *atg11Δ* cells showing delayed cycling of Clb4 levels helps explain unstable aMTs upon loss of Atg11. One of the motor proteins, Kip2, has been identified as a bonafide substrate of Clb4-Cdk1 phosphorylation in regulating aMT stability [95]. Whether Atg11 positively regulates proteins like Kip2 and Stu2 [109], which prevent the catastrophe frequency, or it negatively regulates proteins like Kip3 (the

Kinesin-8 or Kinesin-13 family) or Kar3 which destabilizes MTs *in vivo* [46,109,110], remains to be studied. Furthermore, how a stabilized Clb4 level affects Kar9 phosphorylation is an area of further investigation. It remains to be seen if Atg11 also participates in the maintenance of Kar9 levels. SAC activation leads to the symmetric distribution of Kar9 at the spindle poles [33]. The SAC-mediated delay seen in *atg11Δ* cells, suggests another layer of regulation of Kar9 asymmetry. Taken together, our findings suggest that Atg11 likely participates at multiple pathways in mitosis crucial for accurate chromosome segregation.

Dynamic instability, wherein MTs exhibit cycles of growth and shrinkage is central to high-fidelity chromosome segregation and is therefore tightly regulated during the cell cycle by several proteins and post-translational modifications [46,90,94,109,111–117]. Dynamic MTs facilitate the timely attachment of MTs with the kinetochores in the nucleus and are crucial for the generation of tension between sister chromatids during mitosis [93]. In this study, we demonstrate the significance of Atg11 in maintaining MT integrity, especially under conditions that perturb their rate of polymerization–depolymerization. The presence of Atg11 silences SAC by facilitating chromosome biorientation, leading to securin degradation and timely Cdc14 release by activating the FEAR pathway (Fig 9). SMT assays of Atg11 at the kinetochores do not show specific binding or dynamics at the kinetochores. Therefore, it is unclear how being a cytosolic protein, Atg11 facilitates proper kinetochore-MT attachments within the nucleus during closed mitosis in *S. cerevisiae*. In budding yeast, several proteins regulate MT dynamics both in the nucleus and cytosol. Localized to SPBs, spindles, and the plus-end of aMTs, the MT-binding protein Stu2 is known to regulate kinetochore-MT attachment, aMT, and interpolar MT dynamics [93]. Kar3, a minus-end directed motor, localizes to SPBs, mediates MT depolymerization at SPBs and a loss of Kar3 displays a SAC-dependent delay in the cell cycle [118,119]. SPBs are believed to be a loading platform for MT-associated factors and various cellular components. Whether Atg11 facilitates the dynamics of MTs via Stu2 or Kar3 by being at the SPBs remains an open question.

Our findings are significant in light of the autophagy-independent role of Beclin-1 in accurate kinetochore anchoring to the spindle in cells undergoing open mitosis [120]. Thus, like Beclin-1, Atg11 could be another example of an autophagy-related protein facilitating MT dynamics essential for both kinetochore-MT interactions and spindle positioning during the cell cycle. Depending on the types of mitoses, the players might differ in performing a non-canonical function. Several lines of evidence have established an interplay between errors in the spindle alignment with aging and aneuploidy leading to the development of cancer and neurodegenerative diseases [48–52]. Likewise, perturbations in the autophagy function also contribute to various human diseases like neurodegenerative diseases, cancer, lysosomal storage disorders, and inflammatory disorders [121,122]. Our work adds to the importance of cellular functions carried out by transient protein–protein interactions vital for cellular fitness under specific conditions. Therefore, dissecting these dynamic interactions essential for crosstalk between various biological processes becomes imperative.

## Materials and methods

### Yeast strains, plasmids, and primers

All the strains used in this study are specified in S1 Table. Primer sequences and the plasmids used for strain construction are specified in S2 and S3 Tables, respectively.

### Media and growth conditions

Yeast strains were grown on YPD media (yeast extract 1%, peptone 2%, and dextrose 2%) at 30 °C unless stated otherwise. For the growth assay in the presence of TBZ, overnight grown

cultures of wild-type, autophagy mutants, *ctf19Δ*, *mcm16Δ*, and *ATG11*-complemented strains were serially diluted and spotted on YPD containing 50 µg mL⁻¹ and imaged on 2 dpi or 36 h. The analyses for Figs 3C and 7A–7C were done by growing the cells at 37 °C for 6 h.

## Interaction network

First, we constructed a protein-protein interaction map by identifying the proteins that have been reported to either genetically or physically interact with each of the Atg proteins. We further extended the network to include the proteins that are known to interact with any of the primary interactor proteins. We gleaned these connections using Perl scripts to query the latest curation of the BioGRID yeast interactomes [66]. To this end, only those protein-protein interactions were retained which were confirmed and corroborated across three or more independent studies.

## Construction of yeast strains

The *ATG11* gene was deleted from the *S. cerevisiae* genome by one-step gene replacement [123]. A 40-bp homology region corresponding to the upstream and downstream sequence of *ATG11* having *HPH* marker providing resistance against hygromycin B was amplified from pAG32 using HR34 and HR35 primer pairs. Similarly, *ATG11* was deleted in F2351 [51], with Kar9-GFP and Spc72-mCherry tagged, and in SBY322 [*ipl1-321*, [75]] using primers (HR34 and HR35) carrying 40-bp homology regions from upstream and downstream of the *ATG11* gene that were used to amplify the *HPH* marker sequence from pAG32. The *ATG11* gene deletion in the desired transformants was confirmed by PCR using 5′ and 3′ junction primers (HR36 and HR37).

Double deletion mutants were obtained either by transforming the above *ATG11* deletion cassette in a single deletion mutant background or by crossing a single deletion mutant in *MAT***a** background having the G418 marker with *atg11Δ* cells in *MATα* (BY4742) background having the *HPH* marker. The diploid cells were sporulated and after digestion of the cell wall and vortexing, the cells were plated first on YPD with hygromycin. Growing colonies were patched on G418 and colonies grown on both the selection plates were mated again with both *MAT***a** (BY4741) and *MATα* (BY4742) to differentiate between haploid and diploid cells. Cells showing shmoo formation with only one mating type were taken forward for the assays.

For *atg11Δ* complementation, a ~ 3.9 kb fragment comprising the upstream and full-length sequence of *ATG11* was fused with a ~ 800 bp of *HIS3* fragment with its promoter having a 40-bp homology sequence of *ATG11* at its 3′ end. The purified PCR product was transformed in the *atg11Δ* cells and the desired transformants were confirmed using HR116 and HR117 primers.

## Construction of a strain expressing GFP-Tub1 and Spc42-mCherry under its native promoter

To study the dynamics of MTs, *TUB1* was tagged with GFP using the plasmid, pAFS125 [124]. To visualize SPBs, Spc42, a component of the central plaque of SPB, was tagged with mCherry using the primers HR142 and HR143.

## Construction of *MAD2* deletion strain

The *MAD2* gene was deleted by one-step gene replacement [123], having a 40-bp homology region each from upstream and downstream of the gene using HR52 and HR53 primer pairs. These primers were used to amplify the *HPH* marker from pAG32. The desired deletion strain was confirmed by PCR using 5′ and 3′ junction primers (HR66 and HR67).

## Construction of Pds1-6xHA tagged strain

A PCR fragment containing 6xHA and the *HPH* selection marker was amplified from pYM16 [125] using the primer pair RA140 and RA141. To obtain Pds1-6xHA, the PCR product was transformed in wild-type BY4741 cells generating the strain ScRA403. Transformants were selected using the hygromycin selection marker. To tag Pds1 in *atg11Δ*, *MAT***a** RA403 was crossed with MHR08 (*MATα*). Diploids were selected using methionine and lysine prototrophy. Diploids were sporulated and haploid progeny were identified by mating with both *MAT***a** and *MATα* strains. The desired strain (ScRA405) was confirmed for both *ATG11* deletion and Pds1 tagging using the primer sets HR36 and HR37 and RA146 and RA147, respectively. Pds1-6xHA tagging both in wild-type and *atg11Δ* cells were confirmed by western blot analysis.

## Construction of strains for Bimolecular Fluorescence Complementation (BiFC) assay

We used N-terminal protein tagging vectors, pFA6a-KanMx6-p*GAL1*-VN173 and pFA6a-His3Mx6-p*GAL1*-VC155 for tagging either Atg11 or Spc72 [69]. VN and VC designate N-terminal and C-terminal fragments of Venus, a variant of yellow fluorescent protein, respectively [69]. VN173-Atg11 and VC155-Atg11 were amplified using HR296 and HR297 and HR296 and HR298 primer pairs, respectively. Similarly, VN173-Spc72 and VC155-Spc72 were amplified using HR299 and HR300 and HR299 and HR301 primer pairs, respectively. VN-tagged fragments with the G418 selection and VC-tagged fragments with the *HIS3* selection were transformed into BY4741 and BY4742 backgrounds, respectively. The haploid colonies obtained were mated and selected on G418 plates without histidine to obtain diploid cells expressing both fragments of the fluorophore. The control diploid cells expressing either VN-Spc72 or VC-Atg11 co-expressing Spc42-mCherry were generated by mating BY4741 expressing VN-Spc72 with BY4742 or BY4742 expressing VC-Atg11 with BY4741.

## Construction of strains for Single Molecule Tracking (SMT) assay

The *PDR5* gene, coding for the membrane transporter protein Pdr5 was deleted to allow the retention of the HaloTag ligands (JF646) inside the yeast cell [126]. The *PDR5* gene was deleted in both MHR90 *MAT***a** strain (sfGFP-Atg11 Spc42-mCherry) and yTB283 *MAT***a** strain (sfGFP-Atg11) using a 40-bp homology region each from upstream and downstream of the *PDR5* gene using HR333 and HR334 primer pairs. These primers were used to amplify the *HPH* marker from pAG32. The desired deletion strain was confirmed by PCR using 5′ and 3′ junction primers (HR335 and HR336).

MHR411_a with *PDR5* deletion and sfGFP-Atg11 were transformed with Mtw1-mCherry cassette amplified using RA177 and RA178 primer pairs from pAW8-mCherry plasmid. Mtw1 expressing mCherry was confirmed using fluorescence microscopy. For labeling Atg11 using HaloTag Ligand (HTL), the *ATG11* gene was fused with -HaloTag using the *URA3* selection marker by homologous recombination. The HaloTag cassette was amplified from pTSK561 (Addgene: 190816) using HR329 and HR330 primer pairs and transformed in yTK1501 *MATα* strain. The tagged strains were confirmed by HR36 and HR37 primer pairs. The diploid cells were selected on CM-ura media and YPD supplemented with hygromycin and geneticin.

## Construction of Clb4-9xMyc tagged strain

A PCR fragment comprising 9xMyc and *HIS3* selection marker was amplified from pYM19 [125] using the primer pair HR323 and HR324. The cassette was transformed in both wild-type and *atg11Δ* cells. The transformants were selected on CM-his drop-out media. The transformants expressing Clb4-9xMyc were confirmed by western blot analysis.

## Stability of monocentric plasmid

The *atg1Δ*, *atg6Δ*, *atg11Δ*, *atg15Δ*, *atg17Δ*, *ctf19Δ*, and the isogenic wild-type strains were transformed with pRS313 [127], a centromeric plasmid, and transformants were selected on a dropout media lacking histidine (SD-his). The transformants were then grown in a non-selective media, YPD, for 7–10 generations, followed by plating to obtain single colonies. The single colonies were subsequently patched onto SD-his (selective) and YPD (non-selective) media. The mitotic stability was calculated by the number of colonies that were able to grow on selective media (SD-his) per 100 colonies grown on non-selective media (YPD).

## Chromosome loss assay

The chromosome loss assay was performed in a strain harboring an extra non-essential chromosome that carries *SUP11* that suppresses *ade2* mutation in the assay strain, YMH58a as previously described [128]. Cells retaining this extra chromosome grow as white-colored colonies, complementing the *ade2* mutation. Loss of extra chromosome leads to the red-pigmented colony, displaying *ade2* mutant phenotype, implying loss of *SUP11*. A single white colony of wild-type, *atg11Δ*, and *ctf19Δ* cells were grown in non-selective media (YPD) for 5–7 generations, both at 30 and 37 °C. The cells were plated on YPD. The number of colonies showing chromosome loss at the first division was scored for each condition in three independent biological experiments.

## *In vivo* assay for kinetochore integrity: *CEN* transcriptional read-through assay

The *CEN* transcription read-through assay was done as previously described [73]. Briefly, the reporter plasmid, pAKD06+*CEN4* contains a β-galactosidase reporter gene with the selectable marker *URA3* was transformed in wild-type, *ctf19Δ* and *atg11Δ* cells and selected on SD-ura media. The single colony was inoculated in SD-ura media, containing 2% galactose and 0.3% raffinose and was grown to 1 $OD_{600}$. The cell pellet was washed with water and resuspended in 1 mL of Z-buffer (60 mM $Na_2HPO_4$, 40 mM $NaH_2PO_4.2H_2O$, 10 mM KCl, 5 mM β-ME). 0.1 mL of this was taken for the determination of $OD_{610}$. 0.1 mL of Z-buffer was added to the remaining 0.9 mL of cells. The cells were then permeabilized by adding 50 μL of 0.1% SDS and 100 μL of chloroform (incubated at 30 °C for 15 min). 0.2 mL of 4 mg mL$^{-1}$ ONPG (ortho-nitrophenyl-β-galactoside) was added to the cell suspension and incubated at 30 °C till yellow color developed. The reaction was stopped by adding 0.5 mL of 1 M $Na_2CO_3$. Cells were spun down at 10,000*g* and the clear supernatant was transferred to a fresh tube. The optical density of this solution was measured at 420 and 550 nm. The enzyme activity was normalized with respect to cell density. Units of β galactosidase were measured as:

$$\text{Miller units} = \frac{1000 \left[ \left( OD_{420} \right) - 1.75 \times \left( OD_{550} \right) \right]}{t \times v \times OD_{610}}$$

$t$ = time of reaction $\left( \text{min} \right)$

$v$ = volume of cell suspension used in the assay $\left( \text{mL} \right)$

## Live-cell imaging

For tethering the cells to the glass-bottom dishes for live-cell imaging, the dishes were treated with 6% Concanavalin A (Catalogue no. C2010, Sigma-Aldrich) for 10 min. Wild-type and *atg11Δ* cells carrying GFP-Tub1 (spindle) and Spc42-mCherry (SPBs), or Cdc14-GFP and

Tub1-mCherry, or sfGFP-Atg11 and Spc42-mCherry were grown overnight in CM medium, re-inoculated in CM at 0.2 OD$_{600}$ and grown for 3 h. The cells were first arrested at G1 using 3.0 µg mL$^{-1}$ α-factor at 30 °C (Merck, Cat. No. T6901). After 90 min, 3.0 µg mL$^{-1}$ α-factor was again added and incubated for another 90 min. Cells were washed thrice with CM medium and released into Concanavalin A-treated glass bottom dishes and incubated at 30 °C or 37 °C for 10 min. Live-cell imaging was performed on an inverted confocal microscope (ZEISS, LSM880) equipped with a temperature-controlled chamber (Pecon incubator, XL multiSL), a Plan Apochromat 63× NA oil 1.4 objective and GaAsP photodetectors. For time-lapse micros-copy, images were captured at a 4 min interval with 5% intensity exposure with 0.5 µm Z-steps using 488 and 587 nm for GFP and mCherry as excitation wavelength. All the images dis-played were maximum intensity projections of images (except for Fig 2A, which is represented as a single plane image) for each time using ImageJ.

## Microscopic image acquisition and processing

Wild-type and deletion mutants carrying either GFP-Tub1 (spindle) or Spc42-mCherry (SPB marker) were grown overnight in YPD medium and re-inoculated in YPD at 0.2 OD$_{600}$ for 3 h at 30 °C. The cells were pelleted at 3,000$g$ and mounted on a glass slide having a 2% agarose pad supplemented with 2% dextrose. The pad was sealed with a coverslip. The cells were imaged using an inverted microscope (ZEISS, Axio Observer.Z1/7), a Plan-Apochromat 100×/1.40 oil DIC M27. The filters used were GFP/FITC 488, mCherry 561 for excitation, GFP/FITC 500/550 bandpass or GFP 495/500 and mCherry 565/650 or mCherry 580/750 bandpass for emission. Z-stacks at an interval of 0.5 µm were taken and maximum intensity projection for each image using ImageJ was projected.

## Biorientation assay

For the *CEN3* biorientation assay, overnight grown wild-type and *atg11Δ* cells were reinocu-lated to 0.2 OD$_{600}$ and grown for 3 h. The cells were arrested at metaphase in the presence of 15 µg mL$^{-1}$ of nocodazole for 2 h. The cells were then released in YPD in the presence of 50 µM bathocuproinedisulfonic acid disodium salt (BCS) at 30 or 37 °C for 1 h. Cells with two *CEN3*-GFP puncta between the two SPBs (Spc110-mCherry) were considered bioriented, while those with one *CEN3*-GFP punctum closer to either of the SPBs or away from the polarity axis were considered mono-oriented. The large-budded cells with a budding index of >0.6 were used for the analysis.

For the *CEN3* biorientation assay using live-cell, the overnight grown wild-type and *atg11Δ* cells were re-inoculated to 0.2 OD$_{600}$ and grown for 4 h in CM in the presence of 50 µM BCS and grown at 30 °C. The cells were arrested at metaphase in the presence of 15 µg mL$^{-1}$ of nocodazole for 2 h. The cells were then released in CM in the presence of 50 µM BCS and live-cell was carried out at room temperature. The images were captured at 512×512 frame size with 5 min intervals.

## Western blot for Pds1 dynamics after nocodazole arrest

Overnight grown cells were diluted to 0.1 OD$_{600}$ for wild-type and 0.15 for *atg11Δ* cells in 25 mL complete media. The cultures were then grown at 30 °C until 0.3–0.4 OD$_{600}$. The cells were first arrested at G1 using 2.5 µg mL$^{-1}$ α-factor grown at 25 °C (Cat. No. T6901, Merck). After 90 min, 3.0 µg mL$^{-1}$ α-factor was again added and incubated for another 90 min. Cells were washed with water thrice and released into YPD media containing 15 µg mL$^{-1}$ noco-dazole (Cat. No. M1404, Merck) for 90 min. Metaphase-arrested cells were washed thrice in water and released in 25 mL YPD. The cultures were grown at 30 °C and 1 OD$_{600}$ equivalent

culture was collected and pelleted every 20 min. The cell pellets were lysed by TCA precipitation and resuspended in 50 µL lysis buffer (0.1N NaOH, 1% SDS and 5× protein loading dye). Pds1 degradation dynamics were analyzed using a western blot. Pds1 levels were monitored using rat anti-HA antibody (Cat. No. 11867423001, Roche) and peroxidase-conjugated rabbit anti-rat secondary antibody (Cat. No. HPO14, Genei) at 1:3,000 and 1:10,000 dilution, respectively. GAPDH levels were analyzed using mouse anti-GAPDH antibody (Cat. No. MA5-15738, Invitrogen) and HRP conjugated goat anti-mouse secondary antibody (Cat. No. ab97023, Abcam) at 1:5,000 and 1:10,000 dilution, respectively. The blots were developed using a Chemiluminescence substrate (BioRad) and imaged using the ChemiDoc imaging system (BioRad). Pds1 levels were normalized with their respective loading control at every time point for both wild-type and $atg11\Delta$ cells.

### Western blot for Clb4 dynamics after nocodazole arrest

Overnight grown cells were diluted to 0.2 OD$_{600}$ for wild-type and 0.25 for $atg11\Delta$ cells in 25 mL complete media. The cultures were then grown at 30 °C until 0.4–0.5 OD$_{600}$. The cells were first arrested at G1 using 2.5 µg mL$^{-1}$ α-factor grown at 25 °C (Cat. No. T6901, Merck). After 90 min, 3.0 µg mL$^{-1}$ α-factor was again added and incubated for another 90 min. Cells were washed with complete media thrice and released into YPD media containing 15 µg mL$^{-1}$ nocodazole (Cat. No. M1404, Merck) for 90 min. Metaphase-arrested cells were washed twice in water and once with YPD, and released in 50 mL YPD. The cultures were grown at 30 or 37 °C and 1 OD$_{600}$ equivalent culture was collected and pelleted every 20 min. The cell pellets were lysed by TCA precipitation and resuspended in 80 µL lysis buffer (0.1N NaOH, 1% SDS and 5× protein loading dye). Clb4 degradation dynamics were analyzed using a western blot. Clb4 levels were monitored using mouse anti-Myc antibody (Cat. No. 11867423001, Roche) and peroxidase-conjugated rabbit anti-mouse secondary antibody (Cat. No. ab97023, Abcam) at 1:5,000 and 1:10,000 dilution, respectively. GAPDH levels were analyzed using mouse anti-GAPDH antibody (Cat. No. MA5-15738, Invitrogen) and HRP conjugated goat anti-mouse secondary antibody (Cat. No. ab97023, Abcam) at 1:5,000 and 1:10,000 dilution, respectively. The blots were developed using a Chemiluminescence substrate (BioRad) and imaged using the ChemiDoc imaging system (BioRad). Clb4 levels were normalized with their respective loading control at every time point for both wild-type and $atg11\Delta$ cells.

### GFP-Atg8 processing assay

The $atg1\Delta$, $atg11\Delta$, $atg17\Delta$, and the isogenic wild-type strains were transformed with GFP-Atg8 pRS423, a 2 µm plasmid, and selected on a dropout media lacking histidine (SD-his). The overnight growing cells in SD-his media were reinoculated to 0.25 OD$_{600}$ in SD-his till the OD reached 1.0. The cells were pelleted and washed once with a nitrogen starvation medium (yeast nitrogen base + 2% dextrose). 1 OD cell/mL cells were transferred to a pre-warmed nitrogen starvation medium (30 and 37 °C). 1 mL cells corresponding to 1 OD were aliquoted at 0, 2, 4, and 6 h post incubation in a nitrogen starvation medium. The cell pellets were lysed by TCA precipitation and resuspended in 50 µL lysis buffer (0.1N NaOH, 1% SDS, and 5× protein loading dye). GFP-Atg8 processing assay was analyzed using a western blot. GFP-Atg8 and free GFP levels were monitored using mouse anti-GFP antibody (Roche, Cat. No. 11814460001) and peroxidase-conjugated rabbit anti-mouse secondary antibody (Cat. No. ab97023, Abcam) at 1:6,000 and 1:10,000 dilution, respectively. The blots were developed using a Chemiluminescence substrate (BioRad) and imaged using the ChemiDoc imaging system (BioRad).

## Bimolecular Fluorescence Complementation (BiFC) assay

The overnight growing cells co-expressing Spc72 fused to the N-terminal half of Venus and Atg11 tagged to the C-terminal half of the same fluorophore were reinoculated to 0.2 $OD_{600}$ in YEP containing 2% galactose (YPG) for 3 h. To label the vacuolar membrane, FM4-64 was added at a concentration of 7.5 μM for 30 min. The cells were washed twice with 1× PBS and further incubated for 1 h in YPG before microscopy.

## Staining protocols

FM4-64 staining was performed to visualize the vacuolar membrane. Briefly, FM4-64 (Cat. No. T3166, Thermo Fischer Scientific) was added at a concentration of 7.5 μM for 30 min. The cells were washed twice with 1× PBS and further incubated for 1 h in YPG before microscopy.

Nuclei were visualized by diamidino-2-phenylindole (DAPI) (Cat. No. 62248, Thermo Fischer Scientific) staining. Briefly, the cells were fixed in 3.7% formaldehyde for 10 min. The cells were washed and treated with 0.05% Triton X-100 for 5 min. The cells were resuspended in 1× PBS and stained with 5 ng mL$^{-1}$ DAPI for 10 min and imaged.

Vacuoles were stained with 7-amino-4-chloromethyl-coumarin (CMAC) (ThermoFisher Scientific, Cat. No. C2110; CellTracker Blue CMAC dye) dissolved in DMSO. Cells were incubated with 10 μM dye at room temperature for 90 min, washed with 1× PBS thrice, and were imaged.

## Yeast two-hybrid assays

The full-length *ATG11* (~3,534 bp), *SPC72* (~1,869 bp), and *CNM67* (~1,746 bp) were PCR amplified and cloned in frame with the Gal4 activation/DNA binding domain (AD or BD domain) in pGADC1 and pGBDC1, respectively at BamHI and SalI sites using oligos listed in S2 Table. Yeast-two hybrid analyses were performed in the strain PJ69-4A as described previously [129]. Briefly, bait and prey plasmids were co-transformed into the *S. cerevisiae* strain PJ69-4A. Positive interactions were scored by the appearance of white-colored colonies on a synthetic defined medium containing 6 μg mL$^{-1}$ adenine and/ or by the growth in the presence of 2 mM 3-Amino-1,2,4-triazole (3-AT) (Cat. No. A8056, Sigma-Aldrich).

## aMT dynamics

The overnight grown wild-type and *atg11Δ* cells carrying GFP-Tub1 were reinoculated to 0.2 $OD_{600}$ and were grown for 3 h in CM-ura medium. The cells were first arrested at G1 using 3.0 μg mL$^{-1}$ α-factor grown at 25 °C (Merck, Cat. No. T6901). After 90 min, 3.0 μg mL$^{-1}$ α-factor was again added and incubated for another 90 min. Cells were washed thrice with prewarmed CM-ura medium at 37 °C and released in the same medium at 37 °C till the cells reached metaphase (1.5–2.0 μm). For tethering the cells to the glass-bottom dishes for live-cell imaging, the dishes were treated with 6% Concanavalin A (Cat. No. C2010, Sigma-Aldrich) for 10 min. Cells were resuspended in CM-ura medium and were allowed to adhere for 10 min at 37 °C. The unadhered cells were washed and the adhered cells were resuspended in 3 mL of CM-ura and proceeded for live-cell imaging at 37 °C. The images were acquired every 15 s at 1024×1024 frame size for a total of 25 cycles. At each time point, images were taken at 0.4 μm apart, covering a total depth of 4.5–5.5 μm. Images were processed by a deconvolution process automatically performed by the Zeiss software. Image analysis was done using Fiji. As described in [95,130], various events were defined as follows: (a) catastrophe frequency—the total number of polymerization-to-depolymerization transitions divided by the total time of all growth events; (b) rescue frequency—the total number of depolymerization-to-polymerization transitions divided by the total time of all shrinkage events; (c) polymerization

rate—divide the net change in length by the change in time for each growth event; (d) depolymerization rate—divide the net change in length by the change in time for each shrinkage event. These events were defined only for the bud-directed aMTs and not from aMTs emanating from mSPB.

### Single Molecule Tracking (SMT) assay

The diploid cells tagged with one copy of Atg11 with sfGFP and the other copy tagged with the HaloTag and co-expressing either Spc42-mCherry or Mtw1-mCherry were treated with 40 nM JF646-HTL for 30 min [67]. For carrying out SMT upon TBZ treatment, the cells were first arrested at metaphase in the presence of 200 μg mL$^{-1}$ TBZ for 90 min and then were treated with 40 nM JF646-HTL for 30 min. The cells were then washed thrice for 15 min with either CSM + TBZ or CSM media to remove the unbound JF646-HTL. The cells were then imaged for ~ 1 h using a Leica DMi8 infinity TIRF inverted fluorescence microscope equipped with a Photometric Prime95B sCMOS camera, 100× 1.47 NA TIRF objective lens, 638 nm 150 mW laser module. Time-lapse movies of single molecules were acquired with a 15 ms time interval (total time points collected: 1,000; exposure time: continuous acquisition with 15 ms camera processing time) and a 200 ms time interval (total time points collected: 200; exposure time: 50 ms).

For dwell time analysis, the particle tracking was performed (for 200 ms time interval movies) using the "TrackRecord" software developed in Matlab (The Matworks) [67,131,132]. The software provides automated features for particle detection (using intensity thresholding), and tracking (using the nearest neighbor algorithm with molecules allowed to move a maximum of 6 pixels from 1 frame to the next, and only tracks that are at least 4 frames or longer are kept). Gaps in the tracks due to photoblinking of the fluorescent dye were closed upto 4 frames. The dwell time was determined by fitting the survival distribution [131] after photobleaching correction. Briefly, the survival histogram was generated from the time that each particle was stationary. In practice, even tightly bound particles move slightly due to chromatin and nuclear motion, and therefore a maximum frame-to-frame displacement of 470 nm (Rmin), and a two-frame displacement of 610 nm (Rmax) (both obtained from the motion of the chromatin-bound histone H3 (Hht1, [133]) have been used to define bound portions of each particle's track. Because there is a chance that even a fast-diffusing molecule will move less than these thresholds, a further constraint on the minimum number of time points in the bound segment for each particle (Nmin) is used to reduce <1% the contribution of diffusing molecules to the survival histogram. The Nmin value used for 200 ms time-interval movies was 7. The total bound fraction is then calculated as the ratio of bound track segments to the total number of particles.

To extract residence times, the survival distribution, $S(t)$, is fit by least squares to a mixed exponential decay with two rate constants, $k_{ns} = 1/T_{ns}$ and $k_s = 1/T_s$:

$$S(t) = BX(F_{ns}^{-k_{ns}t} + (1 - F_{ns})^{-k_s t})$$

where $B$ is the bound fraction, and $F_{ns}$ is the fraction of particles non-specifically bound. To check for over-fitting, the distribution is also fit to a single-component exponential:

$$S(t) = B^{-kt}$$

The fits are compared using an $F$-test to ensure that the two-component model gives a significant improvement over the single-component decay.

For quantifying diffusion parameters (fraction of bound/unbound molecules, diffusion coefficient), particle tracking was performed (from 15 ms time-interval movies) using Dia-Track Version 3.05 (Vallotton and Olivier 2013), with the following settings [67]: remove blur 0.02, remove dim 50–250, maximum jump 6 pixels, where each pixel is 110 nm. This software determines the precise position of single molecules by Gaussian intensity fitting and assembles particle trajectories over multiple frames. The trajectory data exported from Diatrack was further converged into a single.csv file using a custom computational package 'Sojourner' (https://rdrr.io/github/sheng-liu/sojourner/). The Spot-On analysis was performed on three frames or longer trajectories using the web interface https://spoton.berkeley.edu/ [68]. The bound fractions and diffusion coefficients were extracted from the cumulative distribution function (CDF) of observed displacements over different time intervals. The cumulative displacement histograms were fitted with a two-state model.

$$p(r,\tau) = F_1 \frac{r}{2(D_1\tau + \sigma^2)} e^{\frac{-r^2}{4(D_1\tau + \sigma^2)}} + Z_{CORR}(\tau, Z, D_2) F_2 \frac{r}{2(D_1\tau + \sigma^2)} e^{\frac{-r^2}{4(D_1\tau + \sigma^2)}}$$

where $F_1$ and $F_2$ are bound and free fractions, $\sigma$ is single molecule localization error, $D_1$ and $D_2$ are diffusion coefficients of bound and free fractions, and $Z_{CORR}$ is the correction factor for fast molecules moving out of axial detection range [68]. The axial detection range for JF646 on our setup is 294 nm. The following settings were used on the Spot-On web interface: bin width 0.01, number of time points 6, jumps to consider 4, use entire trajectories—No, Max jump (μm) 1.5. For model fitting the following parameters were selected: $D_{bound}$ (μm²/s) min 0.0001 max 0.5, $D_{free}$ (μm²/s) min 0.5 max 5, $F_{bound}$ min 0 max 1, Localization error (μm)-Fit from data-Yes min 0.01 max 0.1. Use $Z$ Correction-No, Model Fit CDF, Iterations 3.

## Fluorescence recovery after photobleaching (FRAP) of GFP-tubulin in live-cells

The FRAP assay was carried out as described previously [76]. Briefly, the overnight grown wild-type and *atg11Δ* cells carrying GFP-Tub1 were reinoculated to 0.2 OD$_{600}$ and were grown for 3 h in CM-ura medium. For live-cell imaging, the glass bottom dishes were treated with 6% Concanavalin A (Cat. No. C2010, Sigma-Aldrich) for 10 min. Cells were resuspended in CM-ura medium and were allowed to adhere for 10 min at room temperature. The unadhered cells were washed and the adhered cells were resuspended in 3 mL of CM-ura and proceeded for live-cell imaging at room temperature. The mitotic spindle at the metaphase stage (1.5–2.0 μm) was used for FRAP analysis. Before photobleaching, the images were captured for three-time points at 30 s intervals. One-half of the mitotic spindle was photobleached with a 488 nm laser at 100% intensity with 100 iterations. Images were captured every 30 s at 512×512 frame size at 0.5 μm intervals for a total of 26 cycles. Images were processed by a deconvolution process automatically performed by the Zeiss software. Image analysis was done using Fiji. The images were analyzed as a maximum-intensity projection and quantified as described [76]. A 5-pixelx5-pixel square was placed over the bleached half-spindle, unbleached half-spindle, and a cytoplasmic reference site (for background subtraction) and integrated density was measured for these three sites for every time point. The fluorescence intensity was normalized with the average fluorescence intensity before photobleaching. We calculated the recovery of the bleached region (ratio of bleached to unbleached region after recovery), and the time taken to recover 50% or greater fluorescence intensity of the bleached region after photobleaching. Loss of fluorescence was observed at the unbleached half-spindle, which is in line with the observation that bleached GFP-Tub1 was replaced with the unbleached half-spindle with similar kinetics [76].

## Post-acquisition analysis

Spindle length was measured by tracking it using a straight line, and freehand (for bent spindle) tool in ImageJ after maximum intensity projection for each image. Spindle orientation was measured on images that were maximum intensity projected using the angle tool in ImageJ. The angle was calculated by measuring the smaller of the two angles that the spindle forms intersecting the mother-bud axis of the cell. The aMTs were measured after maximum intensity projection of the images using either a straight line or a freehand tool in ImageJ. The budding index was represented as the ratio of the diameter of the daughter cell to the diameter of the mother cell. Throughout the manuscript, large-budded cells with a budding index of >0.6 are taken into consideration for all the analyses [119]. Statistical analyses were done using GraphPad Prism 8.4.0 software. Student $t$ test followed by Unpaired $t$ test for comparing two groups. The data sets represented as bar graphs were analyzed using either One-way ANOVA or two-way ANOVA for multiple groups and have been mentioned in the figure legends where applicable. Scatter plots with data from three independent biological repeats were plotted as SuperPlots [134], displaying individual color-coded data. The statistical analyses were done with three independent biological replicates.

## Supporting information

**S1 Fig. The *atg11Δ* cells are hyper-sensitive to the microtubule depolymerizing drug thiabendazole (TBZ).** (A) Overnight grown cells of wild-type, null mutants of autophagy-related genes (*atg*), and two kinetochore mutants, *mcm16Δ* and *ctf19Δ*, were 10-fold serially diluted, spotted on YPD and YPD plates containing dimethylformamide (DMF) only or 50 μg mL$^{-1}$ thiabendazole (TBZ). Plates were photographed after incubation at 30 °C for 36 h. (B) Immunoblot analysis of whole-cell lysate prepared at indicated time points from the wild-type, *atg1Δ*, *atg17Δ*, and *atg11Δ* strains expressing GFP-Atg8 grown at 30 and 37 °C in nitrogen starvation medium and probed with anti-GFP antibodies. GFP-Atg8 (38 kDa) and free GFP (25 kDa) are labeled with black and blue arrowheads, respectively. Uncropped western blots are available in S1 Raw Images.
(TIF)

**S2 Fig. Cell cycle-dependent localization of Atg11.** (A) Micrographs showing canonical localization of Atg11 onto the vacuolar membrane during mitosis. The sfGFP-Atg11 expressing cells were co-stained with FM4–64 (staining vacuolar membrane, red) and with CMAC (staining the vacuolar lumen, blue). Scale bar, 5 μm. (B) Micrographs representing different focal planes showing localization of Atg11 (sfGFP-Atg11) closer to the nuclear envelope (Nup49-mCherry, white arrowheads), at various cell cycle stages and co-stained with the vacuolar lumen dye CMAC. Scale bar, 6 μm.
(TIF)

**S3 Fig. Atg11 interacts with Spc72 at the SPBs.** (A) Primary (square boxes) and secondary (oval) interactors of Atg11 as reported in the latest curation of BioGRID [66]. (B) Representative images displaying an *in vivo* BiFC interaction between Atg11 and Spc72 by the reconstitution of Venus fluorescence (VN-Spc72/VC-Atg11) in diploid cells at the vacuolar periphery, marked by FM4–64 staining. Scale bar, 6 μm. (C) Representative images displaying an *in vivo* BiFC interaction between Atg11 and Spc72 at the SPBs, (Spc42-mCherry), in the diploid cells. The white arrowheads mark the co-localization of the BiFC signals with the dSPB. Scale bar, 6 μm. (D) A bar diagram displaying the proportion of cells displaying co-localization of BiFC signals with either dSPB, mSPB or both the SPBs ($n = 85$ large-budded cells). (E) Representative fluorescence images expressing the VN-tagged Spc72 in diploid cells co-expressing Spc42-tagged with mCherry. Scale

bar, 6 μm. (F) Representative fluorescence images expressing VC-tagged Atg11 in diploid cells co-expressing Spc42-tagged with mCherry. Scale bar, 6 μm. (G) Schematic showing the spatial position of proteins at the outer plaque of the SPB. (H) Yeast-two hybrid (Y2H) assays in *S. cerevisiae* strains carrying indicated plasmids were 10-fold serially diluted and spotted on SD - trp - leu or SD - trp - leu - his supplemented with 2 mM 3-amino-1,2,4-triazole (3-AT) and grown at 30 °C. AD- activation domain, BD- DNA binding domain. (I) Yeast-two hybrid (Y2H) assays to study interactions between Atg11 and Cnm67, an outer plaque SPB protein (highlighted in bold in the schematic). The underlying data for panel D can be found in S1 Data.
(TIF)

**S4 Fig. Cells lacking Atg11 display a delayed cell cycle at 30 °C.** (A) Time-lapse images showing dynamics of the mitotic spindle (GFP-Tub1) and SPBs (Spc42-mCherry) during the cell cycle in wild-type (*top*) and *atg11Δ* cells (*bottom*) after G1 arrest followed by release at 30 °C. Scale bar, 5 μm. (B) Scatter plot displaying time taken for wild-type and *atg11Δ* cells, after G1 arrest followed by release at 30 °C, for completion of the cell cycle (*left*), to enter into metaphase (*middle*, 1.5–2 μm mitotic spindle), and anaphase onset till telophase (*right*, disassembly of MTs). Error bars show mean ± SEM. Statistical analysis was done using an unpaired *t* test with Welch's correction (**p = 0.0045/0.0031, ****p < 0.001). (C) A bar diagram representing the proportion of unbudded, small-budded, and large-budded cells in wild-type (WT) and *atg11Δ* cells grown in YPD for 0, 2, 4, and 6 h at 37 °C. More than 100 cells were analyzed for each biological replicate and for every time point, N = 3. Error bars show mean ± SEM. Statistical analysis was done by two-way ANOVA using Tukey's multiple comparisons test (**$p = 0.0068/0.0017$, ****$p < 0.0001$). The underlying data for panels B-C can be found in S1 Data.
(TIF)

**S5 Fig. Stabilized Pds1 levels indicate defects in kinetochore-MT interactions in *atg11Δ* cells.** (A) Schematic (*left*) showing steps involved in synchronizing followed by the release of wild-type and *atg11Δ* cells to study Pds1 protein dynamics after G1 arrest and release at 37 °C. Western blot analysis (*right*) shows the expression of Pds1-6xHA in wild-type and *atg11Δ* cells. Protein levels of GAPDH were used as a loading control. The experiments were repeated twice with similar results. (B) Schematic (*left*) describing the transcriptional readthrough assay [73]. Briefly, the LacZ (blue) coding sequence is in-frame with the amino-terminal actin ORF (black). *CEN* DNA (gray), labeled with I, II, and III represent CDE1, CDEII, and CDEIII, respectively. Transcription is under the control of the *GAL10* promoter and the solid arrow, below the line diagram, represents maximum activity, while the width of dashed arrows represents β-galactosidase activity. A bar diagram (*right*) representing β-galactosidase levels of the corresponding strains. The measurement was performed in triplicates. Statistical analysis was done by one-way ANOVA using Dunnett's multiple comparisons test (p < 0.0001). (C) A bar diagram representing the proportion of cells with bioriented (light blue) or mono-oriented kinetochores (*CEN3*-GFP) (dark gray) in each strain grown at 30 °C. *n* represents the minimum number of large-budded cells (budding index of > 0.6) analyzed in three independent biological replicates. Error bars show mean ± SEM. Statistical analysis was done using two-way ANOVA for multiple comparisons (****$p < 0.0001$). (D) Survival time distribution of Atg11 at kinetochores was quantified from 200 ms time-interval movies. The distribution does not fit well with the double exponential curve, suggesting only a fast fraction. The pie chart represents the percentage of molecules unbound (gray) and bound with short residence time (fast fraction, light green). The average residence time of fast fractions is presented next to their representative fractions. *n* = number of tracks analyzed. The underlying data for panels B-D can be found in S1 Data. Uncropped western blots are available in S1 Raw Images.
(TIF)

**S6 Fig. Cells lacking Atg11 display a delayed Cd14 release**. (A) Time-lapse images showing Cdc14 (Cdc14-GFP) and spindle (Tub1-mCherry) dynamics during the cell cycle in wild-type (*top*) and *atg11Δ* (*bottom*) cells after G1 arrest and release at 37 °C. The yellow arrowheads show Cdc14 release from the nucleolus. Scale bar, 5 μm. (B) The violin plot displays the time duration for wild-type and *atg11Δ* cells to release Cdc14 from the nucleolus, after G1 arrest and release at 37 °C (*top*) or 30 °C (*bottom*). Statistical analysis was done using an unpaired *t* test with Welch's correction (\*$p = 0.0108$, \*\*$p = 0.0014$). The underlying data for panel B can be found in S1 Data. (TIF)

**S7 Fig. The *atg11Δ* cells display an increased number of catastrophe and rescue events, a shorter aMT length at catastrophe and a shorter time span for polymerization**. (A) Bar diagram showing the number of catastrophe events in wild-type and *atg11Δ* cells. (B) Bar diagram displaying aMT length at catastrophe. (C) Bar diagram representing the time taken for polymerization before the catastrophe event. (D) Bar diagram showing the number of rescue events. (E) Bar diagram displaying aMT length at the rescue. (F) Bar diagram representing the time taken for depolymerization before the rescue event. Error bars show mean ± SD. Statistical analysis was done using an unpaired *t* test with Welch's correction (\*\*\*$p = 0.0003$, \*\*\*\*$p < 0.0001$). (G) Scatter plot displaying aMT length at S-phase (<1 μm mitotic spindle) in wild-type (WT) and *atg11Δ* cells grown at 37 °C. (H) Scatter plot representing aMT length at metaphase (1.5–2 μm mitotic spindle) in wild-type (WT) and *atg11Δ* cells grown at 37 °C. (I) Scatter plot showing aMT length at late anaphase (>7 μm mitotic spindle) in wild-type (WT) and *atg11Δ* cells grown at 37 °C. *n*, a minimum number of cells analyzed, $N = 3$. The statistical significance was done using an unpaired *t* test with Welch's correction (\*$p = 0.0194/0.0191$). The underlying data for panels A-I can be found in S1 Data. (TIF)

**S8 Fig. The *atg11Δ* cells exhibit stabilized levels of Clb4.** (A) Schematic (*left*) showing steps involved in synchronizing followed by the release of wild-type and *atg11Δ* cells to study Clb4 protein dynamics after G1 and metaphase arrest and release at 30 °C. Cells were collected every 20 min to prepare protein samples. Western blot analysis (*right*) shows the expression of Clb4-9xMyc in wild-type and *atg11Δ* cells. Protein levels of GAPDH were used as a loading control. Clb4 normalized values are indicated below each lane and the values are plotted as a line graph. The experiments were repeated twice with similar dynamics. The underlying data for panel A can be found in S1 Data. Uncropped western blots are available in S1 Raw Images. (TIF)

**S1 Table. *S. cerevisiae* strains used in this study**. (PDF)

**S2 Table. Oligonucleotides used in this study.** (PDF)

**S3 Table. Plasmids used in this study.** (PDF)

**S1 Data. Data set.** (XLSX)

**S1 Raw Images. Uncropped western blots.** Each of the uncropped western blots is shown and overlaid with the protein marker except for western blots shown in S5A Fig. The lane labeled with X represents protein mraker. (TIF)

## Acknowledgments

We thank J. Heitman, L. Sreekumar, S. Sridhar, V. Yadav, and K. Guin for the critical reading of the manuscript. We thank S. Biggins, D. J. Klionsky, C. Kraft, S. K. Ghosh, S. Bhattacharyya, and F. Monje-Casas for sharing yeast strains and vectors. We acknowledge A. Desai (UCSD, California), G. Dey (EMBL, Heidelberg), G. Pereira (University of Heidelberg), E. Schiebel (DKFZ-ZMBH, Heidelberg), S. Laxman (InStem, Bengaluru) and U. Surana (A*STAR, Singapore) for the valuable suggestions. We thank M. Sirajuddin (InStem, Bengaluru) for the valuable discussion related to aMT dynamics. We also acknowledge B. Bhojappa (IISc, Bengaluru) for providing Cdc14-GFP Tub1-mCherry tagged strain and helping with the imaging and S. Palani (IISc, Bengaluru) for the constructive suggestions. We thank B. Suma and K. Jayaram at the imaging facility at JNCASR. We also thank the members of the Molecular Mycology Laboratory and Autophagy Laboratory for their valuable inputs and comments. Some images were created with BioRender.com.

## Author contributions

**Conceptualization:** Kaustuv Sanyal.

**Data curation:** Md. Hashim Reza, Rashi Aggarwal, Jigyasa Verma, Nitesh Kumar Podh, Ratul Chowdhury.

**Formal analysis:** Md. Hashim Reza, Rashi Aggarwal, Jigyasa Verma, Nitesh Kumar Podh, Gunjan Mehta, Ravi Manjithaya, Kaustuv Sanyal.

**Funding acquisition:** Md. Hashim Reza, Kaustuv Sanyal.

**Investigation:** Md. Hashim Reza, Rashi Aggarwal, Jigyasa Verma, Nitesh Kumar Podh, Gunjan Mehta.

**Methodology:** Md. Hashim Reza, Rashi Aggarwal, Jigyasa Verma.

**Project administration:** Kaustuv Sanyal.

**Resources:** Md. Hashim Reza, Rashi Aggarwal, Jigyasa Verma, Ratul Chowdhury, Gunjan Mehta, Ravi Manjithaya, Kaustuv Sanyal.

**Supervision:** Kaustuv Sanyal.

**Validation:** Md. Hashim Reza, Rashi Aggarwal, Jigyasa Verma, Gunjan Mehta.

**Visualization:** Md. Hashim Reza, Rashi Aggarwal, Jigyasa Verma, Nitesh Kumar Podh.

**Writing – original draft:** Md. Hashim Reza, Rashi Aggarwal, Gunjan Mehta, Kaustuv Sanyal.

**Writing – review & editing:** Md. Hashim Reza, Rashi Aggarwal, Jigyasa Verma, Nitesh Kumar Podh, Ratul Chowdhury, Gunjan Mehta, Ravi Manjithaya, Kaustuv Sanyal.

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
