## [Editor Report · Decision Letter 0]

20 May 2024

Dear Dr Sanyal, 

Thank you for submitting your manuscript entitled "Spindle pole body-associated Atg11, an autophagy-related protein, regulates microtubule dynamics essential for high-fidelity chromosome segregation" for consideration as a Research Article by PLOS Biology.

Your manuscript has now been evaluated by the PLOS Biology editorial staff as well as by an academic editor with relevant expertise and I am writing to let you know that we would like to send your submission out for external peer review.

Once your full submission is complete, your paper will undergo a series of checks in preparation for peer review. After your manuscript has passed the checks it will be sent out for review. To provide the metadata for your submission, please Login to Editorial Manager (https://www.editorialmanager.com/pbiology) within two working days, i.e. by May 22 2024 11:59PM.

Kind regards,

Ines

--

Ines Alvarez-Garcia, PhD

Senior Editor

PLOS Biology

---

## [Decision Letter · Decision Letter 1]

3 Sep 2024

Dear Dr Sanyal,

Thank you for your patience while your manuscript entitled "Spindle pole body-associated Atg11, an autophagy-related protein, regulates microtubule dynamics essential for high-fidelity chromosome segregation" was peer-reviewed at PLOS Biology. Please also accept again my sincere apologies for the delay in providing you with our decision. The manuscript has now been evaluated by the PLOS Biology editors, an Academic Editor with relevant expertise, and by two independent reviewers. 

The reviews are attached below. As you will see, the reviewers find the conclusions potentially interesting and significant for the field, however they also raise several concerns that would need to be addressed before we can consider the manuscript for publication. Reviewer 1 suggests further experiments to confirm the conclusions and to strengthen the model by using live-cell microscopy, and also offers alternative interpretation of the data that should be considered. Reviewer 2 thinks that it should be determined if Spc72 and Atg11 colocalise, and also suggests several experiments to analyse microtubule dynamics and its potential regulation by Atg11.

In light of the reviews, we would like to invite you to revise the work to thoroughly address the reviewers' reports. Given the extent of revision needed, we cannot make a decision about publication until we have seen the revised manuscript and your response to the reviewers' comments. Your revised manuscript is likely to be sent for further evaluation by all or a subset of the reviewers.

**IMPORTANT - SUBMITTING YOUR REVISION**

3. Resubmission Checklist

a) *PLOS Data Policy*

b) *Published Peer Review*

Sincerely,

Ines

--

Ines Alvarez-Garcia, PhD

Senior Editor

PLOS Biology

Reviewers' comments

Rev. 1:

The authors explore whether autophagy-related proteins affect genome stability in budding yeast, Saccharomyces cerevisiae. First they find that Atg11 lacking cells with poor stability of minichromosomes which suggests a segregation inaccuracies. Missegregation can arise from a variety of molecular defects and so they explore these systematically. They find that Atg11 loss leads to an arrest in metaphase, with stabilised securin. They show that Atg11 localise at spindle poles, and its association with Kar9-dependent spindle positioning. Combining these together, the work indicates changes in global microtubules (astral and kinetochore). However, the authors propose a role for changes in dynamic instability leading to segregation accuracy.

The study has a variety of strengths bringing together single-molecule microscopy, genetics and live-cell analysis. The final model could be developed considering all of the data.

The authors present good quality experiments, including live microscopy. This reviewer's main concern is that there is no direct evidence for changes in dynamic instability of kinetochore microtubules, and there is no evidence for checkpoint proteins at the kinetochore. In the absence of such direct evidence the authors can propose biorientation defects as a potential mechanism but not the mechanism for all the cell cycle changes. There could be alternate mechanisms of autophagy related organelle maintenance, protein degradation machineries, stress pathways which could be disrupted. Given the involvement of autophagy in mitotic catastrophe (https://doi.org/10.1038/s41598-017-14901-z), and emerging roles of mitotic autophagy (https://doi.org/10.1242/jcs.255802), Atg11 could be significant for any of these pathways. In fact the authors see changes in a-microtubules more than KT-microtubules. So I suggest either reinterpretation of final conclusion, or exploration for direct evidence such as checkpoint protein recruitment.

Major comments

1) The autophagy protein screen is useful for the community. Of the various components tested, only atg11Δ cells displayed high sensitivity to TBZ but not at high temperatures - this observation could reflect a number of changes that are common to temperature change and microtubule changes (not necessarily changes in microtubule dynamics). (Figure 1)

2) Deletion of MAD2 can appear to rescue any mitotic delay or arrest and this does not directly mean that Atg11 is triggering a checkpoint. This could also be because Atg11 is affecting degradation independent of microtubule - chromosome attachment stability. (Figure 4) The authors themselves see defects in multiple phases of cell cycle (Figure 3)

3) Significant increase in monooriented kinetochores following nocodazole release is an observation that will support the conclusion on failure to biorient (Figure 4D). This is the most significant piece of evidence to support their model and the authors could strengthen it using live-cell microscopy, and also crossing with ipl1 mutants.

4) Single molecule tracking assay could be performed in TBZ and untreated conditions/high temperature to learn about Atg11.

5) Figure 8, they propose a change in astral and nuclear microtubule dynamics (underpinning MT dynamics data needs to be presented). Figure 6 outcomes on MT dynamics are unclear.

Minor point

1) unclear how atg11 localisation at spindle pole is relevant to the paper (Figure-2). It will be useful to see if Atg11 colocalises at the kinetochore or microtubule-ends

2) Experiments on Kar9 and astral microtubule changes do feel more relevant to the phenotype than failure in bioriented attachments

3) The conclusions made in the abstract do not fully match with the interpretations made in the rest of the results (case in point figure 7).

4) Images require scale bars.

5) Figure 4D is unclear

Rev. 2:

Reza et al. investigate the role of Atg11 in yeast chromosome segregation through a series of experiments. They identify Atg11 as a candidate involved in microtubule dynamics by screening for autophagy proteins affecting cell cycle progression under microtubule-depolymerizing conditions. They observe higher chromosome loss and reduced plasmid stability in Atg11-deleted strains, suggesting its role in chromosome segregation. In addition, the authors show that absence of Atg11 leads to an increased in metaphase arrested cells due to activation of the Spindle assembly checkpoint (SAC). Protein interaction analysis reveals that Atg11 interacts with Spc72. By light microscopy the observe that Atg11 transiently localizes near one of the SPB, where it might interact with Spc72. Additional live-cell imaging and single-molecule tracking confirm this interaction. Further studies show that Atg11 depletion disrupts Kar9 localization and microtubule dynamics, causing spindle misorientation and SAC activation. Additionally, Atg11 deletion results in increased microtubule catastrophe frequency and altered rescue dynamics, further emphasizing its effect on microtubule dynamics during cell division. Based on their experiments the authors suggest that Atg11 facilitates microtubule dynamics crucial for chromosome segregation.

Overall, the finding that a protein involved in autophagy affects chromosome segregation and microtubule dynamics is interesting and of relevance. The paper suggests that Atg11 directly affects microtubule dynamics. However, the experiments presented do not convincingly support this conclusion. It might be more plausible that Atg11 affects microtubule dynamics indirectly, possibly by regulating the degradation of cyclins or other proteins that influence the cytoskeleton. The introduction of Atg11 in the manuscript is somewhat lacking, i.e. what is known, what type of protein is Atg11 etc. A more comprehensive background on Atg11's known functions and its role in cellular processes would benefit the reader. Additionally, the citations throughout this section are sparse. Providing more references to foundational and recent studies would strengthen the manuscript's scientific grounding. While the overall study is well-conceived, the authors should revisit some of their interpretations, particularly those that propose direct effects where indirect mechanisms might be more plausible. Additional experiments could be suggested to clarify the role of Atg11 in the observed phenomena.

Major comments:

1) The authors Identify Spc72, one of the components of the spindle pole body (SPB), as a potential interactor by protein-protein interaction network analysis. Live-cell imaging (Fig 2A) reveals a transient and proximal localization of Atg11with on of the SPBs (dSPB) just prior to anaphase.

a) It would be very informative if the authors could provide a little bit more detail how the proposed co-localization of Spc72 and Atg11 was determined. Was this "quantified" by eye or was a co-localization analysis performed. The images in Figure 2A show "co-localization" at 40' and 56' but only 40' is counted as a co-localization, could the authors explain how they "discarded" 56' as not co-localizing?

b) According to the Material and Methods the images are maximum intensity projections of z-stacks, those could be misleading in terms of co-localization as proteins might look like they are co-localizing but could indeed be in very different z-positions. Would it be possible to show co-localization in a single plane?

c) Previous literature reported the localization of Atg11 as localizing to the pre-autophagosomal structure (PAS). This localization looks very similar to the localization shown here but is not discussed at all. Do the authors also observe localization to the PAS? How can this be distinguished.

2) The authors report that Atg11 depletion leads to "more random" Kar9 localization. The authors should discuss the possible effect of erroneous Kar9 localization. I think three aspects are highly relevant:

a) Kar9 plays a crucial role in spindle orientation during cell division, particularly in asymmetric cell divisions. Kar9 guides the positioning of the mitotic spindle by interacting with microtubules and motor proteins, ensuring that the spindle aligns properly along the mother-bud axis in budding yeast. In addition, Kar9 was suggested to bind microtubules as well as interact with BIM1, the yeast homolog of the microtubule plus-end binding protein EB1. This could suggest that the effect on Microtubule dynamics is driven by changes in Kar9 localization upon Atg11 depletion, rather than a direct effect of Atg11 on microtubule dynamics. This could be addressed in the discussion.

b) Kar9 is regulated through a degradation pathway that starts at kinetochores. This process requires Kar9 to be near the SUMO-targeted ubiquitin ligases Slx5-Slx8, which is facilitated by a transport of Kar9 into the nucleus and its recruitment to kinetochores via SUMO. If this degradation pathway is disrupted, Kar9 accumulates on cytoplasmic microtubules (cMTs), leading to spindle mislocalization into the bud. Additionally, the buildup of Kar9 on cMTs causes Bim1 to be retained in the cytoplasm, resulting in errors in chromosome segregation.

This suggests that possibly the degradation of Kar9 could be affected in the Atg11 depleted cells.

c) The localization and function of Kar9 are tightly regulated by phosphorylation events, particularly by the cyclin-dependent kinase Cdc28 (CDK1 in humans). Phosphorylation affects Kar9's association with microtubules, influencing its asymmetric localization to the SPB and function during mitosis. Interestingly, both CDKs and cyclins are also targets of autophagy.

Along this line, a different study by Zucca et al 2023 (PMID 36269172) found that astral microtubule (aMT) dynamics are carefully controlled. They found that aMTs stay unstable until metaphase and then stabilize at the start of anaphase. This change in aMT behavior, crucial for proper spindle orientation, depends on the degradation of the mitotic cyclin Clb4 by the Anaphase Promoting Complex (APC) with its activator subunit Cdc20 (APC/CCdc20). These findings reveal a specific role for Clb4 in regulating aMT.

Is it possible that Atg11 affects the degradation of cyclins and thus Kar9 and spindles? Does the phosphorylation of Kar9 change in absence of Atg11 possibly hinting towards a role of Atg11 in degradation of cyclins?

The Authors could use western blots to detect changes in phosporylation levels of Kar9 as well as degradation of cyclins.

3) Are microtubule dynamics changed globally in the entire cells or locally (SPB specific)? Or rather, given that Atg11 localization is asymmetric, are microtubule dynamics on the SPB with Atg11 different from the one without?

Microtubule dynamics could be quantified locally.

4) Based on their experiments the authors suggest that "Atg11 facilitates microtubule dynamics crucial for chromosome segregation". The data clearly shows that cells depleted of Atg11 have altered microtubule dynamics, however the data does not allow the conclusion that Atg11 directly regulates microtubule dynamics. The discussion of the data should reflect this.

Minor Comments:

Figure 7B label for atg11 depleted is missing

Those two lines contradict each other in terms of effects on rescue frequency. A significant increase in rescue events should also lead to an increase in rescue frequency. However, authors here report no difference in rescue frequency:

p28, line 477: " The atg11Δ cells displayed a significantly increased catastrophe frequency (Fig. 6C), while the rescue frequencies were comparable to those of the wild-type"

p29, line 493:" The atg11Δ cells also showed a significant increase in the number of rescue events"

---

## [Decision Letter · Decision Letter 2]

5 Jan 2025

Dear Dr Sanyal,

Thank you for your patience while we considered your revised manuscript entitled "Autophagy-related protein Atg11 is essential for microtubule-mediated process of chromosome segregation" for publication as a Research Article at PLOS Biology. This revised version of your manuscript has been evaluated by the PLOS Biology editors, the Academic Editor and two of the original reviewers.

Based on the reviews, we are likely to accept this manuscript for publication, provided you satisfactorily address the data and other policy-related requests stated below.

In addition, we would like you to consider a suggestion to improve the title:

"Autophagy-related protein Atg11 is essential for microtubule-mediated chromosome segregation"

We expect to receive your revised manuscript within two weeks. 

*Published Peer Review History*

*Press*

Sincerely,

Ines

--

Ines Alvarez-Garcia, PhD

Senior Editor

PLOS Biology

Fig. 1C, D; Fig. 2E; Fig. 3B, C; Fig. 4A, C, D, F, H; Fig. 5A, B; Fig. 6C-F; Fig. 7A-D; Fig. 8A, C; Fig. S3D; Fig. S4B, C; Fig. S5B-D; Fig. S6B; Fig. S7A-I and Fig. S8A

CODE POLICY

Reviewers' comments

Rev. 1:

The point by point response letter is well articulated and the authors have addressed all my queries satisfactorily.

The model has been strengthened by new evidence on a delay in cell cycle progression (slow growth) in ATG11 and IPL1 double mutant.

The new Nocodazole release experiment is also a very valuable addition to this paper and the field.

I am happy to recommend the publication of the manuscript.

Rev. 2:

I would like to thank the authors for addressing the reviewers comments. I think this significantly improved this manuscript and I have no further comments.

---

## [Editor Report · Decision Letter 3]

13 Feb 2025

Dear Dr Sanyal,

Thank you for the submission of your revised Research Article entitled "Autophagy-related protein Atg11 is essential for microtubule-mediated chromosome segregation" for publication in PLOS Biology. On behalf of my colleagues and the Academic Editor, Jonathon Pines, I am delighted to let you know that we can in principle accept your manuscript for publication, provided you address any remaining formatting and reporting issues. These will be detailed in an email you should receive within 2-3 business days from our colleagues in the journal operations team; no action is required from you until then. Please note that we will not be able to formally accept your manuscript and schedule it for publication until you have completed any requested changes.

PRESS

Sincerely, 

Ines

--

Ines Alvarez-Garcia, PhD

Senior Editor

PLOS Biology
